# PostEdit: Posterior Sampling for Efficient Zero-Shot Image Editing

**Feng Tian**[1], **Yixuan Li**[1], **Yichao Yan**[1], **Shanyan Guan**[2], **Yanhao Ge**[2], **Xiaokang Yang**[1,*]

[1]MoE Key Lab of Artificial Intelligence, AI Institute, Shanghai Jiao Tong University
[2]vivo Mobile Communication Co., Ltd
`{tf1021, lyx0208, yanyichao, xkyang}@sjtu.edu.cn`
`{guanshanyan, halege}@vivo.com`

## Abstract

In the field of image editing, three core challenges persist: controllability, background preservation, and efficiency. Inversion-based methods rely on time-consuming optimization to preserve the features of the initial images, which results in low efficiency due to the requirement for extensive network inference. Conversely, inversion-free methods lack theoretical support for background similarity, as they circumvent the issue of maintaining initial features to achieve efficiency. As a consequence, none of these methods can achieve both high efficiency and background consistency. To tackle the challenges and the aforementioned disadvantages, we introduce PostEdit, a method that incorporates a posterior scheme to govern the diffusion sampling process. Specifically, a corresponding measurement term related to both the initial features and Langevin dynamics is introduced to optimize the estimated image generated by the given target prompt. Extensive experimental results indicate that the proposed PostEdit achieves state-of-the-art editing performance while accurately preserving unedited regions. Furthermore, the method is both inversion- and training-free, necessitating approximately 1.5 seconds and 18 GB of GPU memory to generate high-quality results. Code: https://github.com/TFNTF/PostEdit.

## 1 Introduction

Large text-to-image diffusion models Saharia et al. (2022); Pernias et al. (2024); Podell et al. (2024); Ramesh et al. (2022) have demonstrated significant capabilities in generating photorealistic images based on given textual prompts, facilitating both the creation and editing of real images. Current research Cao et al. (2023); Brack et al. (2024); Ju et al. (2024); Parmar et al. (2023); Wu & la Torre (2022); Xu et al. (2024) highlights three main challenges in image editing: *controllability*, *background preservation*, and *efficiency*. Specifically, the edited parts must align with the target prompt's concepts, while unedited regions should remain unchanged. Moreover, the editing process must be sufficiently efficient to support interactive applications. As illustrated in Fig. 1, there are two mainstream categories of image editing approaches, namely inversion-based and inversion-free methods.

Inversion-based approaches Song et al. (2021a); Mokady et al. (2023); Wu & la Torre (2022); Huberman-Spiegelglas et al. (2024) first invert a clean image to a noisy latent (*inversion phase*) and then denoising the latent conditioned on the given target prompt to obtain the edited image (*editing phase*). However, directly inverting the diffusion sampling process inevitably introduces deviations with the input image, due to error accumulated by the unconditional score term (discussed in classifier-free guidance (CFG) Ho & Salimans (2022) and proven in App. A.14). Consequently, the editing quality of inversion-based methods is primarily constrained by the similarity in unedited regions. Several approaches address this issue by optimizing the text embedding Wu et al. (2023), employing iterative guidance Kim et al. (2022); Garibi et al. (2024), or directly modifying attention layers Hertz et al. (2023); Mokady et al. (2023); Parmar et al. (2023) to mitigate the bias introduced by the unconditional term. However, the necessity of adding and subsequently removing noise predicted by a network remains unavoidable, thereby significantly constraining their efficiency. Recent

---

*corresponding author: xkyang@sjtu.edu.cn

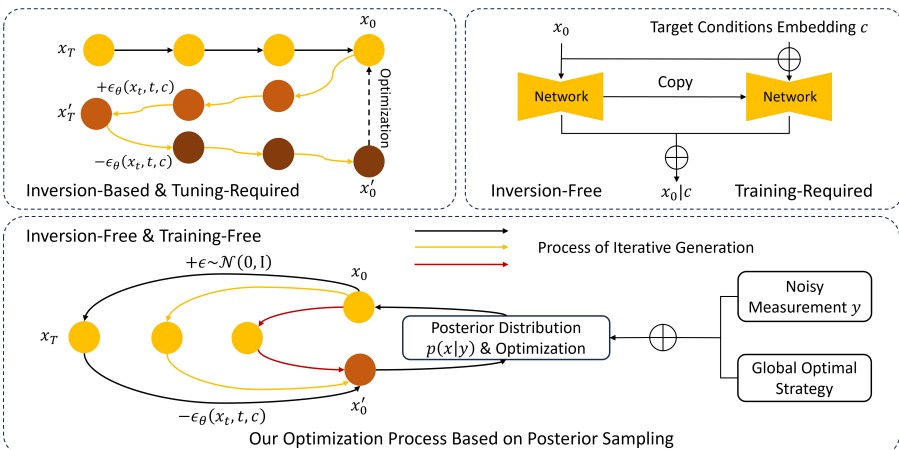

Figure 1: **Different Image Editing Schemes.** The *inversion-based* method, illustrated in the top-left section, involves adding noise from a pre-trained network to a clean image. It then denoises the image based on a target prompt, though it requires time-consuming tuning to ensure background preservation. The top-right section discusses *training-based, inversion-free* methods, which train a learnable model to achieve satisfactory results but have limited generalization capabilities. Our approach, outlined in the bottom section, is both *inversion-free and training-free*.

methods Starodubcev et al. (2024); Li & He (2024); Kim et al. (2024) attempt to enhance the accuracy of the iterative sampling process by training an invertible consistency trajectory, following the distillation process in the consistency models (CM) Song et al. (2023); Salimans & Ho (2022); Song & Dhariwal (2024); Luo et al. (2023b). Although this approach significantly reduces the accumulation errors from the unconditional term, it cannot eliminate them. Moreover, the editing performance is sensitive to the hyperparameters (*i.e.*, the fixed boundary timesteps of multi-step consistency models), and the training process generally demands hundreds of GPU hours.

Another category of methods Brooks et al. (2023); Mou et al. (2024); Ye et al. (2023); Guo et al. (2024); Li et al. (2023); Wang et al. (2024) is inversion-free and thus significantly decreases the inference time. The general idea is to train networks to learn to embed the given conditions into the noisy-to-image diffusion process. For example, ControlNet Zhang et al. (2023b) and T2I-Adapter Mou et al. (2024) train an extra network to encode the image-shaped conditions, *e.g.*, depth maps, canny maps. However, these works highly rely on the accuracy of the input guidance structure, while most applications related to ControlNet involve customization. Some other works Zhang et al. (2023a), Zhang et al. (2024b), Hui et al. (2024) employ a diffusion model trained on synthetic edited images, producing edited images in a supervised manner. This methodology obviates the need for inversion process during the sampling stage. Moreover, there is a training-free method to satisfy inversion-free requirement Xu et al. (2024). It adopts specific settings of the DDIM solver to leverage the advantages of CM to ensure the editing quality. Although these recent works can achieve fast sampling and accurate editing, the aforementioned problem remains unsolved since the diffusion sampling process Ho et al. (2020); Song et al. (2021a;b) is necessary. Therefore, all the inversion-free methods cannot circumvent the accumulation errors caused by the unconditional score term in CFG.

In this work, we present an inversion- and training-free method named PostEdit to optimize the accumulated errors of the unconditional term in CFG based on the theory of posterior sampling Kawar et al. (2021; 2022); Chung et al. (2023); Zhang et al. (2024a;a); Lugmayr et al. (2022); Zhu et al. (2023); Song et al. (2021b). To reconstruct and edit an image $x_0$, we adopt a measurement term $y$ which contains the features of the initial image, and supervise the editing process by the posterior log-likelihood density $\nabla_{x_t} \log p(x_t|y)$. With this term, we can estimate the target image through progressively sampling from the posterior $p(x_t|y)$ referring to the Bayes rule. The above process is reasonable since the inverse problems of probabilistic generative models are ubiquitous in generating tasks, which are trained to learn scores to match gradients of noised data distribution (log density), and this process is also called score matching Song & Ermon (2020), Song & Ermon (2019), Karras et al. (2022) and Karras et al. (2024). $y$ is defined according to the following inverse problem

$$y = \mathcal{A}(x_0) + n, \tag{1}$$

where $\mathcal{A}$ is a forward measurement operator that can be linear or nonlinear and $\boldsymbol{n}$ is an independent noise. Hence, the posterior sampling strategy can be regarded as a diffusion solver and it can edit images while maintaining the regions that are required to remain unchanged with the measurement $\boldsymbol{y}$. Also, instead of time-consuming training or optimization, our framework adopts an optimization process without requirements for across the network many times for inference, which can be lightweight taking about $1.5$ seconds to operate and around $18$ GB of GPU memory. Our contributions and key takeaways are shown as follows:

- To the best of our knowledge, we are the first to propose a framework that extends the theory of posterior sampling to text-guided image editing task.

- We theoretically address the error accumulation problem by introducing posterior sampling, and designing an inversion-free and training-free strategy to preserve initial features. Furthermore, we replace the step-wise sampling process with a highly efficient optimization procedure, thereby significantly accelerating the overall sampling process.

- PostEdit ranks among the fastest zero-shot image editing methods, achieving execution times of less than 2 seconds. Additionally, the state-of-the-art CLIP similarity scores on the PIE benchmark attest to the high editing quality of our method.

## 2 PRELIMINARIES

### 2.1 SCORE-BASED DIFFUSION MODELS

We follow the continuous diffusion trajectory Song et al. (2021b) to sample the estimated initial image $\hat{\boldsymbol{x}}_0$. Specifically, the forward diffusion process can be modeled as the solution to an Itô SDE:

$$d\boldsymbol{x} = \boldsymbol{f}_t(\boldsymbol{x})dt + g_t d\boldsymbol{w}, \tag{2}$$

where $\boldsymbol{f}$ is defined as the drift function and $g$ denotes the coefficient of noise term. Furthermore, the corresponding reverse form of Eq. 2 can be written as

$$d\boldsymbol{x} = \left[ \boldsymbol{f}_t(\boldsymbol{x}) - g_t^2 \nabla_{\boldsymbol{x}} \log p_t(\boldsymbol{x}) \right] dt + g_t d\bar{\boldsymbol{w}}, \tag{3}$$

where $\bar{\boldsymbol{w}}$ represents the standard Brownian motion. As shown in Song et al. (2021b), there exists a corresponding deterministic process whose trajectories share the same marginal probability densities as the SDE according to Eq. 2. This deterministic process satisfies an ODE

$$d\boldsymbol{x} = \left( \boldsymbol{f}_t(\boldsymbol{x}) - \frac{1}{2} g_t^2 \nabla_{\boldsymbol{x}} \log p_t(\boldsymbol{x}) \right) dt. \tag{4}$$

The ODE can compute the exact likelihood of any input data by leveraging the connection to neural ODEs Chen et al. (2018). To approximate the log density of noised data distribution $\nabla_{\boldsymbol{x}} \log p_t(\boldsymbol{x})$ at each sampling step, a network $\boldsymbol{s}_{\boldsymbol{\theta}}(\boldsymbol{x}_t, t)$ is trained to learn the corresponding log density

$$\mathbb{E}_{\boldsymbol{x}_0, \boldsymbol{x}_t \sim p(\boldsymbol{x}_t | \boldsymbol{x}_0)} \left[ \| \boldsymbol{s}_{\boldsymbol{\theta}}(\boldsymbol{x}_t, t) - \nabla_{\boldsymbol{x}_t} \log p(\boldsymbol{x}_t | \boldsymbol{x}_0) \|^2 \right]. \tag{5}$$

### 2.2 DDIM SOLVER AND CONSISTENCY MODELS

The DDIM solver is widely applied in training large text-to-image diffusion models. The iterative scheme for sampling the previous step is defined as follows

$$\boldsymbol{x}_{t-1} = \sqrt{\alpha_{t-1}} \left( \frac{\boldsymbol{x}_t - \sqrt{1 - \alpha_t} \boldsymbol{\epsilon}_{\boldsymbol{\theta}}(\boldsymbol{x}_t, t)}{\sqrt{\alpha_t}} \right) + \sqrt{1 - \alpha_{t-1}} \boldsymbol{\epsilon}_{\boldsymbol{\theta}}(\boldsymbol{x}_t, t), \tag{6}$$

where $\boldsymbol{\epsilon}_{\boldsymbol{\theta}}(\boldsymbol{x}_t, t)$ is the predicted noise from the network. According to Eq. 6, the sampling process can be regarded as first estimating a clean image $\boldsymbol{x}_0$, and then using the forward process of the diffusion models with noise predicted by the network to the previous step $\boldsymbol{x}_{t-1}$. Therefore, the predicted original sample $\hat{\boldsymbol{x}}_0$ is defined as

$$\hat{\boldsymbol{x}}_0 = \frac{\boldsymbol{x}_t - \sqrt{1 - \alpha_t} \boldsymbol{\epsilon}_{\boldsymbol{\theta}}(\boldsymbol{x}_t, t)}{\sqrt{\alpha_t}}. \tag{7}$$

Latent consistency models Luo et al. (2023a) apply the DDIM solver Song et al. (2021a) to predict $\hat{x}_0$ and use the self-consistency of an ODE trajectory Song et al. (2023) to distill steps. Then the $x_0$ is calculated by the function $f_{\theta}(z, c, t)$ through large timestep, where $f$ is defined in Eq. 7

$$f_{\theta}(z, c, t) = c_{\text{skip}}(t)z + c_{\text{out}}(t)\left(\frac{z - \sigma_t \epsilon_{\theta}(z, c, t)}{\alpha_t}\right), \tag{8}$$

$z$ is denoted as $x$ encoded in the latent space. The loss function of self-consistency is defined as

$$\mathcal{L}_{\mathcal{CD}}\left(\theta, \theta^-; \Psi\right) = \mathbb{E}_{z, c, n}\left[d\left(f_{\theta}\left(z_{t_{n+1}}, c, t_{n+1}\right), f_{\theta^-}\left(\hat{z}_{t_n}^{\Psi}, c, t_n\right)\right)\right], \tag{9}$$

where $\hat{z}_{t_n}^{\Psi}$ is an estimation of the evolution of the $z_{t_n}$ from $t_{n+1}$ using ODE solver $\Psi$.

## 2.3 POSTERIOR SAMPLING IN DIFFUSION MODELS

After obtaining $s_{\theta}(x_t, t)$, we can infer an unknown $x \in \mathbb{R}^d$ through the degraded measurement $y \in \mathbb{R}^n$. Specifically, in the forward process, it is well-posed since the mapping $x \to y : \mathbb{R}^d \to \mathbb{R}^n$ is many-to-one, while it is ill-posed for the reverse process since it is one-to-many when sampling the posterior $p(x_0|y)$, where it can not be formulated as a functional relationship. To deal with this problem, the Bayes rule is applied to the log density terms and we can derive that

$$\nabla_{x_t} \log p(x_t|y) = \nabla_{x_t} \log p(x_t) + \nabla_{x_t} \log p(y|x_t), \tag{10}$$

where the first term in the right side hand of the equation is the pre-trained diffusion model and the second one is intractable. The measurement $y$ can be regarded as a vital term that contains the information of the prior $p(x)$, which supervises the generation process towards the input images. In order to work out the explicit expression of the second term, existing method DPS Chung et al. (2023) presents the following approximation

$$p(y|x_t) = \mathbb{E}_{x_0 \sim p(x_0|x_t)}[p(y|x_0)] \approx p(y|\hat{x}_0), \quad \hat{x}_0 = \mathbb{E}_{x_0 \sim p(x_0|x_t)}[x_0], \tag{11}$$

where the Bayes optimal posterior $\hat{x}_0$ can be obtained from a given pre-trained diffusion models or Tweedie's approach to iterative descent gradient for the case of VP-SDE or DDPM sampling. Hence, each step can be written as $p(x_{t-1}|x_t, y)$ according to Eq. 10.

When the transition kernel is defined, since the solvers utilize the unconditional scores to estimate $\hat{x}_0$, the measurement term is then introduced through a gradient descent way to optimize $x$

$$x_{t-1} = f(x_t, \hat{x}_0, \epsilon) + \eta \nabla_{x_t} \|y - \mathcal{A}(\hat{x}_0)\|_2^2, \quad \epsilon \sim \mathcal{N}(0, I), \tag{12}$$

where the function $f$ is defined as the approximation of the unconditional counterpart of $p(x_{t-1}|x_t, y)$ and $\eta$ denotes the learning rate.

## 3 METHOD

We propose a novel sampling process integrated with a tailored optimization procedure that incorporates the measurements $y$ and Langevin dynamics to enhance the quality of image reconstruction and editing. The adopted SDE/ODE solver is based on DDIM, as described in Eq. 6. Denote $z \sim \mathcal{E}(x_0)$, $z \in \mathbb{R}^p$ where $\mathcal{E}$ is an encoder and $x_0$ is an initial image. Our method operates in latent space and leverages the theory of the posterior sampling to correct the bias from the initial features and introduce the target concepts. The core insight is using $y$ in the form of Gaussian distribution, estimated $z_0$ and Langevin dynamics as the optimization terms to correct the errors of the sampling process. The importance of reconstruction and the algorithm are introduced specifically in (Sec. 3.1). The implementation details of the editing process are illustrated in detail (Sec. 3.2). PostEdit takes around 1.5 seconds and 18 GB memory costs on a single NVIDIA A100 GPU.

## 3.1 POSTERIOR SAMPLING FOR IMAGE RECONSTRUCTION

The quality of reconstruction is a crucial indicator for evaluating the editing capabilities of a method. To preserve the features of the background (areas unaffected by the target prompt), Mokady et al. (2023) introduces a technique for fine-tuning the text embedding to mitigate errors caused by the

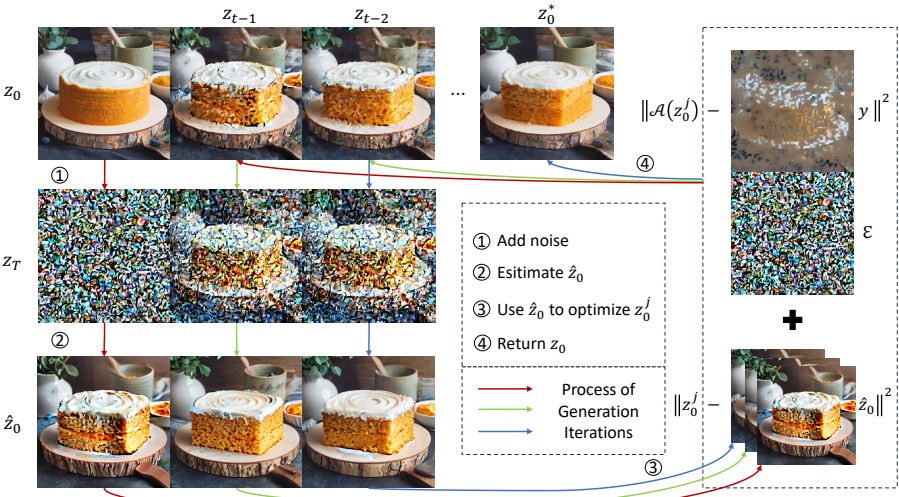

Figure 2: **Method Overview.** The latent representation of initial image $x_0$ is $z_0$. It is adding noise randomly to $z_T$ and then $\hat{z}_0$ is estimated from $z_T$ through diffusion ODE solvers. After that, there are two optimization terms relating to $\hat{z}_0$, the given measurement $y$ and a random noise term $\epsilon$, which is applied to optimize calculated $\hat{z}_0$ while avoids solutions falling in local optimality. Then the optimized $\hat{z}_0$ is adding noise to timestep $T-1$ according to the noise scheduler. This process operates recursively and finished till $\hat{z}_T$ is converged to $z_0$, where $z_0^*$ is the finally optimized output.

null text term, as demonstrated in App. A.14. However, this approach is time-consuming, and there is a pressing need to enhance editing performance. To address this challenge, we propose a method that enables a fast and accurate reconstruction and editing process.

Specifically, there are four steps in our method: (1) We add noise to $z_0$ following the DDPM noise schedule until $z_T \sim \mathcal{N}(\mathbf{0}, \mathbf{I})$. Unlike the iterative inversion process used in Mokady et al. (2023) (DDIM inversion), where multiple network inferences are required, the added noise here is directly sampled from $\mathcal{N}(\mathbf{0}, \mathbf{I})$. As a result, this process adds random noise directly to the clean image in a single step, significantly reducing the computational time. (2) Existing SDE/ODE solvers, such as DDIM and LCM, are employed to estimate $\hat{z}_0$. (3) To ensure that $\hat{z}_0$ aligns more consistently with the background and target prompt features of the original image, it is optimized using two $L_2$ norm terms related to the defined measurement $y$ and $\hat{z}_0$ respectively. Additionally, Langevin dynamics is employed to avoid convergence to local optima. (4) Finally, by progressively applying the above process to update the mean and variance of the Gaussian distribution in the Step (1) according to a predefined time schedule, we obtain $z_0^*$ with consistent initial features and accurate target characteristics respectively as $T$ converges to $0$. The presented algorithm corresponding to the above process is shown in detail in Fig. 2 for image reconstruction or editing and Alg. A.15 for image reconstruction. The measurement $y$, defined in Eq. 17, can be simplified as a masked observation. For a deeper understanding, we recommend referring to Kawar et al. (2021).

Our method requires a large text-to-image diffusion model as input and we select Stable Diffusion (SD) Rombach et al. (2022). Given that SD is trained on the dataset containing billions of images, the generated result has strong randomness relating to the same prompt. Therefore, if we directly apply posterior sampling strategy shown in Eq. 12 to acquire $z_0$ from $z_T \sim \mathcal{N}(\mathbf{0}, \mathbf{I})$ by leveraging SD to inference noise, $\hat{z}_0$ differs greatly from the ground truth $z_0$. Moreover, reconstructing an image with a given text prompt starting from $\mathcal{N}(\mathbf{0}, \mathbf{I})$ usually gets poor results for SD due to the bias caused by the unconditional term in CFG. Conversely, posterior sampling has good reconstruction performance when it leverages the diffusion model trained on small datasets, for example, FFHQ Karras et al. (2019) and ImageNet Deng et al. (2009) as shown in Chung et al. (2023); Zhang et al. (2024a). We experimentally discover that the gap between these two kinds of models is the inconsistent layouts of each estimated image $\hat{z}_0$, while the features of the target prompts are successfully introduced into the generated $\hat{z}_0$. Specifically, the layouts of $\hat{z}_0$ generated by the scores inferred by the networks trained on FFHQ and Laion-5B Schuhmann et al. (2022) for intermediate timesteps are shown in App. A.4. Therefore, due to the editing and reconstruction trade-off issue, it is much more challenging for high-quality image editing and reconstruction by leveraging large text-to-image models.

To address the editing and reconstruction trade-off issue, we present a weighted process that introduces the features of initial data into the estimated $\hat{z}_0$ as shown in the following proposition.

**Proposition 1.** *The weighted relationship between the estimated $\hat{z}_0$ and the initial image $z_{in}$ to correct evaluated $z_0$ is defined as* $(0 \leq w \leq 0.1)$

$$z_0^w = (1-w) \cdot \hat{z}_0 + w \cdot z_{in}, \tag{13}$$

*where $w$ is a constant to govern the intensity of the injected features.*

By additionally introducing $z_{in}$ (weighted by $w$) during sampling, we can produce more high-fidelity and similar layout with the input image, which is critical for applying posterior sampling for image reconstruction and editing.

*Remark* 1. Eq. 13 is reasonable since $(1-w)$ and $w \cdot z_{in}$ are regarded as constant. Hence, this process does not essentially influence the sampling process of distribution $z_{t-1} \sim \mathbb{E}\left(\mathcal{N}\left(z_0^w, \sigma_{t-1}^2 \boldsymbol{I}\right)\right)$, which is shown specifically in the Proposition 2.

In order to adapt Eq. 10 to the DDIM solver shown in Eq. 6 adopted by SD, we can derive it as

$$\nabla_{z_0} \log p(z_0|z_t, \boldsymbol{y}) = \nabla_{z_0} \log p(z_0|z_t) + \nabla_{z_0} \log p(\boldsymbol{y}|z_0, z_t), \tag{14}$$

to calculate the scores towards to $z_0$ straightly inspired by Chung et al. (2023) and Zhang et al. (2024a). The measurement settings for image reconstruction are listed in the App. A.2

**Proposition 2.** *Suppose $z_t$ is sampled from time marginal distribution of $p\left(z_t|\boldsymbol{y}\right)$, then*

$$z_{t-1} \sim \mathbb{E}_{z_0^w} \mathcal{N}\left(z_0^w, \sigma_{t-1}^2 \boldsymbol{I}\right), \tag{15}$$

*satisfies the time marginal distribution conditioned on $p\left(z_{t-1}|\boldsymbol{y}\right)$, where $z_0^w$ is obtained from Eq. 13. (Proof is shown in Appendix A.16)*

Propsition 2 ensures that $z_{t-1}$ sampled from the Gaussian distribution (with mean $z_0^w$ and variance $\sigma_{t-1}^2$) still satisfies the constraint of the posterior sampling Eq. 10. Therefore, we can present the following scheme to optimize the estimated $\hat{z}_0$ and run Langevin dynamics Welling & Teh (2011):

$$z_0^{(k+1)} = (1-w) \cdot z_0^{(k)} + w \cdot z_{in} - h \cdot \nabla_{z_0^{(k)}} \left( \frac{\|z_0^{(k)} - z_0\|^2}{2\sigma_t^2} + \frac{\|\mathcal{A}\left(z_0^{(k)}\right) - \boldsymbol{y}\|^2}{2m^2} \right) + \sqrt{2h}\boldsymbol{\epsilon}. \tag{16}$$

Here, $\mathcal{A}\left(\cdot\right)$ is identical to $\boldsymbol{P}\left(\cdot\right)$ as defined in Eq. 17 and $h$ is the step size. $\sigma_t$ and $m$ are hyperparameters detailed in Appendix A.2. Additionally, $\boldsymbol{\epsilon} \sim \mathcal{N}(\boldsymbol{0}, \boldsymbol{I})$. Fig. 15 and Fig. 16 in Appendix A.10 demonstrate that PostEdit can achieve high-quality reconstruction outcomes without requiring any tuning process. Eq. 16 is reasonable since the two terms that multiplied by the step size $h$ have the same descent direction towards to the ground truth $z_0$. Additionally, Langevin dynamics is employed to search for solutions that achieve a global optimum. Since the Step (1) shown in Fig. 2 is different from the process of DDIM inversion from the initial image to noise, which involves adding noise $\boldsymbol{\epsilon}_\theta\left(z_t, t, c_{ini}\right)$ inferred by the network at each step Mokady et al. (2023) (where $c_{ini}$ represents the prompt describing the content of the initial image), PostEdit is much more efficient as mentioned before. Considering that no information related to the initial image is incorporated into the noised distribution, the term involving the measurement $\boldsymbol{y}$, as defined in Eq. 16, is introduced to correct errors in the initial features caused by the unconditional term in CFG. This ensures background consistency. All parameter settings are detailed in Appendix A.2. The detailed process of image reconstruction is outlined in Alg. 2 of Appendix A.15.

## 3.2 POSTERIOR SAMPLING FOR IMAGE EDITING

In this section, we present details of the posterior sampling process for high-quality *image editing* using the DDIM solver Luo et al. (2023a), as outlined in Eq. 6. Unlike the ODE solver used in the image reconstruction task described in Sec. 3.1, image editing requires the solver with higher accuracy to estimate $\hat{z}_0$. The measurement $\boldsymbol{y}$ for image reconstruction and editing is defined as

$$\boldsymbol{y} \sim \mathcal{N}\left(\boldsymbol{P}z, \sigma^2 \boldsymbol{I}\right), \tag{17}$$

---

**Algorithm 1** Posterior Sampling for Image Editing

---

1: **Require:** Diffusion model $\epsilon_\theta$, step size $h$, posterior sampling steps $N$, diffusion solver steps $n$, image $x_0$, measurement $y$, weight $w$, target prompt $c_{tgt}$ , coefficients of diffusion sampler $c_{skip}$ and $c_{out}$, encoder $\mathcal{E}$, decoder $\mathcal{D}$, noise schedule $\alpha(t)$, $\sigma(t)$, posterior sampler sequence $\{\tau_i\}_{i=0}^{N-1}$ and diffusion sampler sequence $\{t_j\}_{j=0}^{n-1}$.

2:   $z_0 \sim \mathcal{E}(x_0)$, $z_{in} = z_0$

3: **for** $i = N - 1$ to $0$ **do**

4:     **for** $j = n - 1$ to $0$ **do**

5:       Sample $z_j \sim \mathcal{N}\left(\alpha\left(t_j\right) z_0, \sigma^2\left(t_j\right) I\right)$

6:       $z_0 = c_{\text{skip}}\left(t\right)z_j + c_{\text{out}}\left(t\right)\left(\frac{z_j - \sigma_t \epsilon_\theta(z_j, c_{tgt}, t)}{\alpha_t}\right)$

7:     **end for**

8:     $z_0^0 = z_0$

9:     **for** $k = 0$ to $T - 1$ **do**

10:       Sample $\epsilon \sim \mathcal{N}(0, I)$.

11:       $z_0^{(k+1)} = (1 - w) \cdot z_0^{(k)} + w \cdot z_{in} - h \cdot \nabla_{z_0^{(k)}}\left(\frac{\|z_0^{(k)} - z_0\|^2}{2\sigma_t^2} + \frac{\|\mathcal{A}\left(z_0^{(k)}\right) - y\|^2}{2m^2}\right) + \sqrt{2h}\epsilon$

12:     **end for**

13:     Sample $z_{\tau_{i-1}} \sim \mathcal{N}(z_0^{(T)}, \sigma_{\tau_{i-1}}^2 I)$.

14:     $z_0 = c_{\text{skip}}\left(t\right)z_{\tau_{i-1}} + c_{\text{out}}\left(t\right)\left(\frac{z_{\tau_{i-1}} - \sigma_t \epsilon_\theta(z_{\tau_{i-1}}, c_{tgt}, t)}{\alpha_t}\right)$

15: **end for**

16: $x_0 = \mathcal{D}\left(z_0\right)$

17: **Return** $x_0$

---

where $P \in \{0, 1\}^{n \times p}$ represents a masking matrix composed of elementary unit vectors. This measurement setup not only serves as a specialized configuration for image editing but also demonstrates its capacity to deliver high-quality image reconstruction results, even when $z_0$ is masked. The settings in Eq. 17 have been validated to yield high-quality reconstructions, as shown in Sec. 4.3, demonstrating the method's effectiveness in preserving the features of the initial image.

Furthermore, to minimize the number of sampling steps, improving the accuracy of the estimated $\hat{z}_0$ is crucial. According to the experimental results presented in LCM, the superior denoising capabilities of the LCM solver Luo et al. (2023a) are demonstrated to surpass those of the DDIM solver in both speed and accuracy. Consequently, we utilize the LCM solver, distilled from models based on the DDIM solver, to markedly improve the convergence rate and produce more accurate $\hat{z}_0$ estimates that closely align with the target prompt in fewer than four steps. The measurement characteristics $y$, as defined in Eq. 17, involve randomly masking each element of $z_0$ with a given probability. Since one of the optimization terms focuses on only a small portion of the initial image, both terms in Eq. 16 guide the gradient descent in the same direction. As the sampling process progresses, the edited $x_{tgt}$ gradually inherits features from both $x_0$ and the target prompt by selectively replacing the necessary attributes. The experimental results of different settings for the optimization defined in Eq. 16 are presented in Sec. 4.4. The rest of the process mirrors the reconstruction phase, allowing us to progressively achieve the edited $x_0$. In summary, the algorithm's procedure is detailed in Alg. 1, with implementation specifics provided in Appendix A.2.

## 4 EXPERIMENTS

### 4.1 EXPERIMENT SETUP

To ensure a fair comparison, all experiments were conducted on the PIE-Bench dataset Ju et al. (2024) using the same parameter settings specified in Appendix A.2 and a single A100 GPU to evaluate both image quality and inference efficiency. The PIE-Bench dataset comprises 700 images with 10 types of editing, where each image is paired with a source prompt and a target prompt. In our experiments, the resolution of all test images was set to $512 \times 512$. For the reconstruction experiments, we set the initial and target prompts to be identical across all test runs. Additional settings, including forward operators, are provided in the Appendix A.2.

| Method | Background Preservation | | | | Efficiency |
|---|---|---|---|---|---|
| | PSNR$^\uparrow$ | LPIPS$^\downarrow_{\times 10^2}$ | MSE$^\downarrow_{\times 10^3}$ | SSIM$^\uparrow_{\times 10^2}$ | Time$^\downarrow$ |
| NTI | **25.58** | **7.98** | **4.37** | **77.02** | $\sim$120s |
| NPI | 24.66 | 9.11 | 4.73 | 76.14 | $\sim$15s |
| iCD | 19.64 | 17.13 | 13.50 | 66.48 | $\sim$1.8s |
| DDCM | 18.00 | 17.74 | 18.94 | 64.01 | $\sim$2s |
| Ours | 24.39 | 9.00 | 4.75 | 72.74 | $\sim$**1.5s** |

Table 1: **Quantitative Comparisons of Image Reconstruction.** All of the comparison methods include strategies specifically designed for image reconstruction.

| Method | Background Preservation | | | | CLIP Similarity | | Efficiency |
|---|---|---|---|---|---|---|---|
| | PSNR$^\uparrow$ | LPIPS$^\downarrow_{\times 10^2}$ | MSE$^\downarrow_{\times 10^3}$ | SSIM$^\uparrow_{\times 10^2}$ | Whole$^\uparrow$ | Edited$^\uparrow$ | Time$^\downarrow$ |
| NTI | 27.50 | 5.67 | 3.40 | 85.03 | 25.08 | 21.36 | $\sim$120s |
| NPI | 25.81 | 7.48 | 4.34 | 83.44 | 25.52 | 22.24 | $\sim$15s |
| PnP | 22.31 | 11.29 | 8.31 | 79.61 | 25.92 | 22.65 | $\sim$240s |
| DI | 27.28 | 5.38 | 3.25 | 85.34 | 25.71 | 22.17 | $\sim$60s |
| iCD | 22.80 | 10.30 | 7.96 | 79.44 | 25.61 | 22.33 | $\sim$1.8s |
| DDCM | 28.08 | 5.61 | 7.06 | 85.26 | 26.07 | 22.09 | $\sim$2s |
| TurboEdit | 22.44 | 10.36 | 9.51 | 80.15 | 26.29 | 23.05 | $\sim$**1.2s**$^*$ |
| SPD | **28.86** | **3.42** | **2.33** | **86.86** | 25.54 | 21.50 | $\sim$30s |
| GR | 25.03 | 7.29 | 4.71 | 83.34 | 25.83 | 22.43 | $\sim$30s |
| IP2P | 19.65 | 17.99 | 26.26 | 75.19 | 24.93 | 21.71 | $\sim$10s |
| OmniGen | 19.63 | 15.06 | 38.76 | 72.29 | 25.18 | 21.77 | $\sim$70s |
| SeedX | 18.79 | 17.52 | 20.82 | 74.93 | 25.76 | 22.34 | $\sim$7s |
| Ours | 27.04 | 6.38 | 3.24 | 82.20 | **26.76** | **24.14** | $\sim$1.5s |

Table 2: **Quantitative Comparisons of Image Editing.** '$*$' indicates models that benefit from SDXL-Turbo's improved inference.

## 4.2 QUANTITATIVE COMPARISON

**Image Reconstruction.** The methods have special design for image reconstruction are compared: NTI Mokady et al. (2023), NPI Miyake et al. (2023), iCD Starodubcev et al. (2024) and DDCM Xu et al. (2024). The results of quantitative comparison are shown in Tab. 1. Although NTI and NPI achieve better performance on the listed metrics, their computational time costs are substantially higher, exceeding ours by at least an order of magnitude. Compared to the highly efficient inversion-free method, DDCM, PostEdit demonstrates significantly superior performance.

**Image Editing.** We compare our method against recent inversion-based and training-based image editing approaches: NTI, NPI, PnP Tumanyan et al. (2023), DI Ju et al. (2024), iCD, DDCM, TurboEdit Deutch et al. (2024), SPD Li et al. (2024) and GR Titov et al. (2024), IP2P Brooks et al. (2023), OmniGen Xiao et al. (2024) and SeedX Ge et al. (2024)[1]. The comparison is evaluated from three aspects: background consistency, CLIP Radford et al. (2021) similarity, and efficiency. The experimental results shown in Tab. 2 reflect that PostEdit achieves SOTA performance on editing, which are the "Whole" and "Edited" metrics of the CLIP similarity and the results are significantly better than others. For efficiency, our model is highly efficient with a runtime less than 2 seconds. It is worth noting that our runtime is slightly higher than TurboEdit Deutch et al. (2024), which is mainly due to different baselines. Specifically, TurboEdit employs SDXL-Turbo Sauer et al. (2023) while our framework is based on LCM-SD1.5 Luo et al. (2023a). As shown in Appendix A.11, SDXL-Turbo Sauer et al. (2023) is almost 2.5 times faster than LCM-SD1.5 Luo et al. (2023a). We believe the efficiency of our framework can be further improved if we adopt a more efficient baseline like SDXL-Turbo. In terms of background preservation, our method achieves the best MSE result among all methods with a runtime of less than 2 seconds. The following section presents additional qualitative results, further illustrating the superiority of our framework in editing capabilities and background preservation, while maintaining high efficiency.

## 4.3 QUALITATIVE COMPARISON OF RECONSTRUCTION AND EDITING

**Image Reconstruction.** We present results of qualitative comparison in Fig. 3. The experiments indicate that PostEdit demonstrates greater robustness and high quality generation ability compared to NTI Mokady et al. (2023), NPI Miyake et al. (2023) and iCD Starodubcev et al. (2024). Specifically, we compared the reconstruction quality across four distinct categories of images: single-object

---

[1]See Appendix A.12 to get instructions for IP2P, OmniGen, and SeedX based on the input and target prompt.

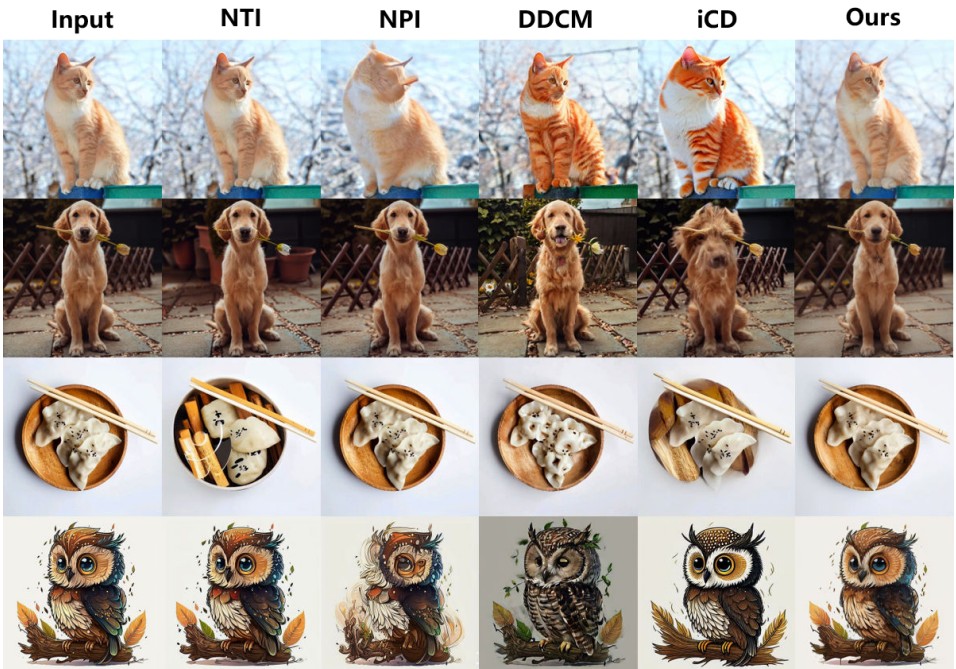

Figure 3: **Qualitative Comparison of Reconstruction.** It takes 1.5 seconds for our method to reconstruct the input image, and the time is 1.8s, 2s, 15s, and 120s for iCD, DDCM, NPI, and NTI, respectively. Our framework can faithfully reconstruct the foreground object and the background.

images, complex backgrounds, multi-object scenes, and cartoon images. The inversion-free method DDCM Xu et al. (2024) fails to faithfully reconstruct the input images, supporting our claim made in Sec. 1. While other methods yield better results in the given cases, they require significantly longer processing times to achieve competitive outcomes. Therefore, our approach offers the best overall performance when considering inference efficiency, stability in generation, and image quality. More reconstruction results on complex objects are shown in Fig. 15 and Fig. 16.

**Image Editing.** The qualitative comparisons of the image editing results are shown in Fig. 4. The effects of text insertion, deletion, and substitution are provided. PostEdit effectively highlights the features present in the target prompt, which aligns with the quantitative results shown in Tab. 2. To present these findings more clearly, we selected the best-performing classical methods, and their results are shown in Tab. 2. For a comparison of the other baseline methods, please refer to Fig. 12 in Appendix A.7. Additionally, the visualized experiments demonstrate that our method successfully preserves the original features. More comparison results can be found in Appendix A.7.

### 4.4    ABLATION STUDY

In this section, we conduct various ablation studies and present the results to demonstrate the effectiveness of our framework. (a) We remove the optimization component shown in Eq. 16 and directly apply the adopted SDE/ODE solver to estimate $x_0$. The experimental results indicate that the edited images lack background preservation. For instance, in the slanted bicycle example shown in the first row of Fig. 5, the staircase on the left side of the original image is transformed into a car in the edited image. (b) We modify the masked probability of our measurement $y$. Notably, there is no discernible difference between the edited images and the input images. (c) We investigate the influence of Proposition 1 on the experimental outcomes, which highlights the effectiveness of the parameter $w$ concerning background similarities. Additionally, the quantitative results, as detailed in Tab. 5, highlight the adopted configurations of PostEdit achieve optimal generation performance.

## 5    CONCLUSION AND LIMITATION

In this work, we address the errors caused by the unconditional term in Classifier-Free Guidance by introducing the theory of posterior sampling to enhance reconstruction quality for image editing.

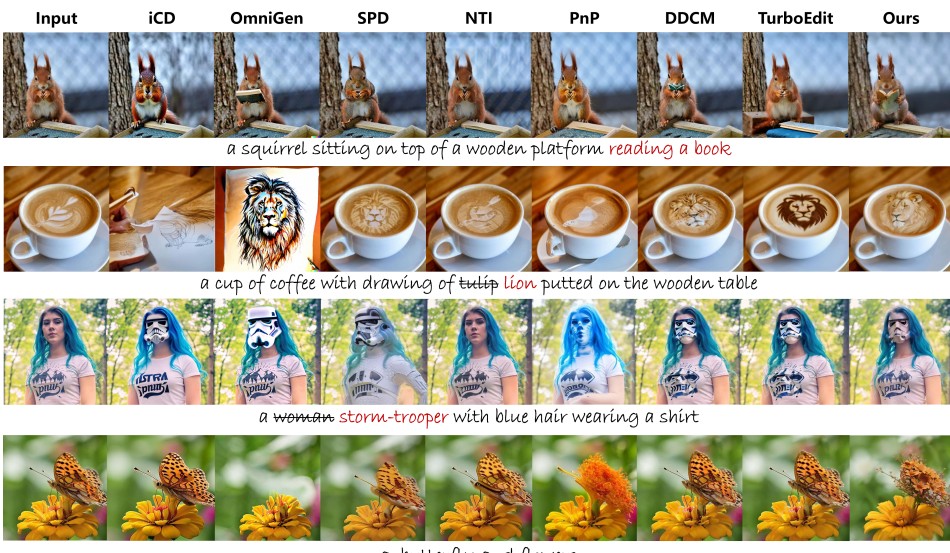

Figure 4: **Qualitative Comparison of Editing.** Our method performs better than the others in aligning with target prompts while maintaining the background similarity.

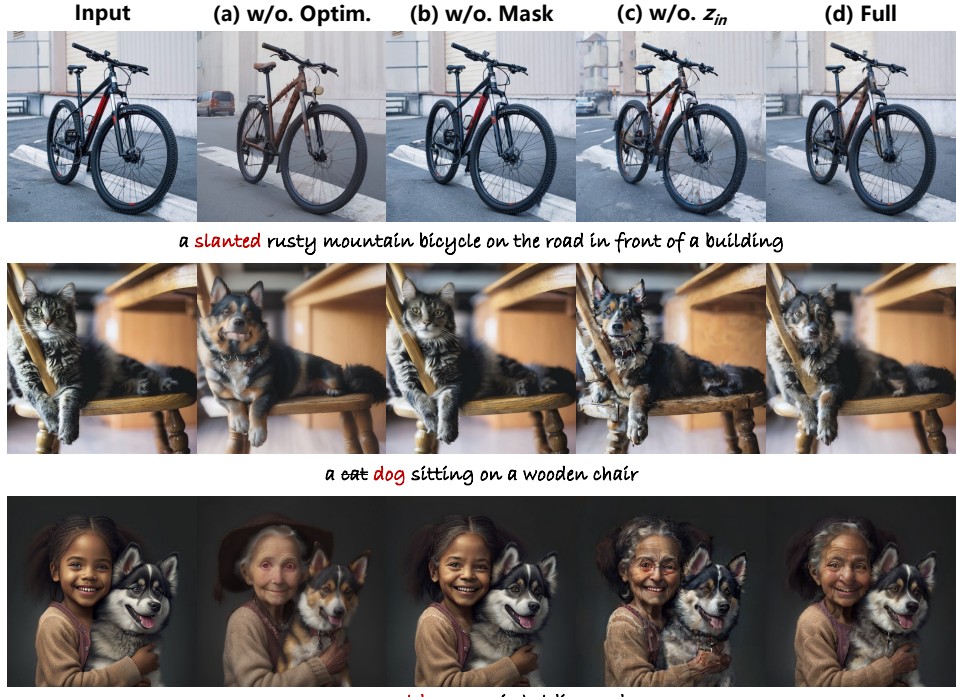

Figure 5: **Ablation Studies.** We show the results without the optimization process shown in Eq. 16, the measurement $y$ defined in Eq. 17 and $z_{in}$ shown in Proposition 1.

By minimizing the need for repeated network inference, our method demonstrates fast and accurate performance while effectively preserving background similarity, as evidenced by the results. Ultimately, our approach tackles three key challenges associated with image editing and showcases state-of-the-art performance in terms of editing capabilities and inference speed.

**Limitation:** PostEdit faces challenges in representing highly specific scenes. For example, describing "a man raising his hand" is considerably more difficult compared to the structured input formats used in ControlNet-related methods. Furthermore, its ability to maintain background consistency is limited and requires improvement. Additionally, the quality and speed of generation are strongly influenced by the performance of the underlying baseline models.

## 6 ACKNOWLEDGEMENTS

This work was supported in part by NSFC (62201342), and Shanghai Municipal Science and Technology Major Project (2021SHZDZX0102). Authors would like to appreciate the Student Innovation Center of SJTU for providing GPUs.

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

# A APPENDIX

## A.1 RELATED WORK IN IMAGE EDITING

Previous works in image editing can be broadly categorized into two paradigms: *inversion-based* and *training-based*.

**Inversion-based Methods.** Several works Zhang et al. (2023d); Ruiz et al. (2023); Gal et al. (2023) focus on modifying the training process of diffusion models to incorporate the information from the initial images. Specifically, the optimization process can be operated in two different spaces: textual space and model space. In textual space, a set of methodologies Dong et al. (2022); Valevski et al. (2023); Han et al. (2023) aim to optimize textual embeddings to perform various editing tasks effectively. In model space, a research line Ruiz et al. (2023); Qiao et al. (2024); Avrahami et al. (2023) intends to further updates modules from the base model to enhance reconstruction capabilities. For example, several studies Kawar et al. (2023); Shi et al. (2024); Zhang et al. (2023c) optimize both textual embeddings and model parameters to ensure content consistency following non-rigid editing or localized distortions. For forward-based inversion, it can also be divided into two categories, which is DDIM inversion and DDPM inversion. Previous works like Mokady et al. (2023) and Miyake et al. (2023) are designed to approximate the inversion trajectory to deal with

accumulated errors. Dong et al. (2023) focuses on optimizing the text embedding, which is then interpolated with the target embedding during the editing process. Some approaches Han et al. (2024); Cho et al. (2024) aim to bypass the time-consuming optimization processes of the aforementioned methods while preserving their reconstruction capabilities. Inspired by normalizing flow models Dinh et al. (2015; 2017), EDICT Wallace et al. (2023) reformulates DDIM processes by simultaneously tracking two associated noisy variables at each step during inversion. These variables can be exactly derived from one another during the sampling phase.

**Training-based Methods.** Since some advanced approaches in zero-shot or few-shot settings require time-consuming optimization Ruiz et al. (2023); Gal et al. (2023); Mokady et al. (2023) or are highly sensitive to hyperparameters Cao et al. (2023); Kawar et al. (2023); Hertz et al. (2023); Huberman-Spiegelglas et al. (2024), several studies Brooks et al. (2023); Ye et al. (2023) aim to train task-specific models with substantial amounts of data to directly transform source images into target images under user guidance. Instruction-based editing Brooks et al. (2023); Zhang et al. (2023a); Xie et al. (2023) provides an intuitive approach for image manipulation, allowing users to input command-style text instead of providing an exhaustive description. For image inpainting, a group of methods Huang et al. (2024); Wang et al. (2023) focuses on completing missing parts of an image under text guidance. Additionally, image translation Isola et al. (2017); Zhang et al. (2023b) seeks to transform the source image into a target domain, such as converting night to daytime or sketch to a natural image. Another type of training-based method is content-free editing, which Ruiz et al. (2023); Wei et al. (2023) aims to preserve the high-level semantics of the source images in the final results. Content-free editing can be further categorized into subject-driven customization and attribute-driven customization. Subject-driven customization Wei et al. (2023); Li et al. (2023); Arar et al. (2023); Chen et al. (2023) is designed to capture the identity of the target and generate novel images that place it in new contexts. In contrast, attribute-driven customization Lee et al. (2024) focuses on extracting and manipulating attributes in a more fine-grained manner.

## A.2 IMPLEMENTATION DETAILS

The main hyper-parameters of the PostEdit are briefly summarized in Tab. 3.

**SD Model.** We adopt LCM-SD1.5 for all the experiments Luo et al. (2023a).

**Parameters of Consistency models.** $c_{skip}$ and $c_{out}$ shown in line 6 of Alg. 1 are set to 0 and 1 for most cases respectively.

**Hyper-parameters in Alg. 1.** $N$ is set to 5 for schedule $\{\tau_i\}_{i=0}^{N-1}$. To ensure higher efficiency and quality at the same time, $z_N$ is sampled through

$$z_N \sim \mathcal{N}\left(\sqrt{\bar{\alpha}_t}z_0, \sqrt{1-\bar{\alpha}_t}I\right), \tag{18}$$

where $t$ is set to 501 generally following the DDPM scheduler Ho et al. (2020). Additionally, $n$ is set to 1 for the sequence of diffusion sampler $\{t_j\}_{j=0}^{n-1}$ to further improve inference speed. The parameter $T$ shown in line 16 of Alg. 1 is set to 100 to ensure optimal quality. For Eq. 16, $m$ is set as 0.01 for both the reconstruction and editing task while $\sigma_t$ corresponds to the timestep of the DDPM scheduler. Generally, we apply Eq. 8 for 1 step to estimate $z_0$, and then according to the following schedule to make $z_{\tau_i}$ progressively converge to $z_0$.

$$\{\tau_i\}_{i=1}^5 = \{501, 401, 301, 201, 101, 1\}. \tag{19}$$

The parameter $w$ is usually set to a minimal value such as 0.1 for most cases or 0 and 0.2 for easy and hard cases.Additionally, $h$ is initially set to 1e-5 for image editing and reconstruction tasks. It is dynamically adjusted at each recursion step, as described in lines 9 to 12 of Alg. 1, using the following equation:

$$h = \left(1 + \frac{k}{T} \cdot (0.01 - 1)\right) \cdot h, \tag{20}$$

where $k$ and $T$ are the same with definition of line 9 of Alg. 1.

**ODE Solvers.** We adopt the solver of LCM Luo et al. (2023a) distilled from Dreamshaper v7 fine-tune of Stable-Diffusion v1-5 for images editing task. For reconstruction, different solvers, for instance, DDIM Song et al. (2021a), DDPM Ho et al. (2020), and Song et al. (2023) based on Stable Diffusion Rombach et al. (2022) are able to work out satisfied reconstruction quality.

| Notation | Values | Description |
|---|---|---|
| $f$ | 0.5 | Appearance probability of 0 in Matrix $P$ shown in Eq. 17 |
| Optimization Steps | 100 | Operating steps of Eq. 16 |
| $w$ | 0.1 | The weighting coefficient of Proposition. 1 |

Table 3: **Main Hyper-parameters.**

| Method | Background Preservation | | | | CLIP Similarity | | Efficiency |
|---|---|---|---|---|---|---|---|
| | PSNR$^\uparrow$ | LPIPS$^\downarrow_{\times 10^2}$ | MSE$^\downarrow_{\times 10^3}$ | SSIM$^\uparrow_{\times 10^2}$ | Whole$^\uparrow$ | Edited$^\uparrow$ | Time$^\downarrow$ |
| $f = 0.3$ | 27.20 | 6.09 | 2.91 | 82.77 | 25.93 | 22.40 | ∼1.5s |
| $f = 0.7$ | 24.43 | 12.16 | 6.06 | 77.64 | 26.73 | 24.28 | ∼1.5s |
| Optimization Steps $= 50$ | 25.49 | 9.39 | 4.85 | 79.95 | 26.61 | 23.59 | ∼1.1s |
| Optimization Steps $= 150$ | 26.59 | 7.19 | 3.77 | 82.05 | 26.51 | 23.47 | ∼1.8s |
| $w = 0.3$ | 27.00 | 8.50 | 4.39 | 82.70 | 26.34 | 23.49 | ∼1.5s |
| $w = 0.5$ | 27.75 | 5.47 | 2.84 | 83.61 | 25.83 | 22.19 | ∼1.5s |
| $w = 0$ | 24.01 | 10.92 | 6.24 | 77.28 | 26.45 | 23.45 | ∼1.5s |
| Ours Default | 27.04 | 6.38 | 3.24 | 82.20 | 26.76 | 24.14 | ∼1.5s |

Table 4: **Quantitative Results of Hyperparameter Sensitivity Analysis**

**Probability of Masked Features.** We use the probability equal to 0.5 for a randomly mask process, which represents whether one of the latent features is masked or not.

**Measurement $y$ Used for Better Quality of Image Reconstruction.** The measurement $y$ can be chosen from the following Eq. 21 to further improve the reconstruction ability of PostEdit, which are defined as linear and nonlinear operations relating to initial image $z_0$ in latent space

$$y \sim \mathcal{N}\left(|\boldsymbol{FPz}_0|, \sigma^2 \boldsymbol{I}\right), \tag{21}$$

where $\boldsymbol{F}$ and $\boldsymbol{P}$ denote the 2D discrete Fourier transform matrix and the oversampling matrix with ratio $k/n$ respectively for Eq. 21. However, the forward operator term shown in Eq. 21 reflects poor editing ability, and all our editing and reconstruction results all based on the measurement shown in Eq. 17.

**Oversampling Matrix.** We set $\sigma$ shown in Eq. 21 to 0.01 and use an oversampling factor $k = 2$ and $n = 8$.

**2D Discrete Fourier Transform Matrix.** The 2D Fourier transform is defined as

$$F[u, v] = \frac{1}{\sqrt{MN}} \sum_{x=0}^{M-1} \sum_{y=0}^{N-1} f(x, y) \exp\left[-j 2\pi \left(\frac{xu}{M} + \frac{yv}{N}\right)\right],$$

$$u = 0, 1, ..., M-1; \quad v = 0, 1, ..., N-1, \tag{22}$$

where $f(x, y)$ is denoted as a two-dimensional discrete signal with dimension $M \times N$ obtained by sampling at superior intervals in the spatial domain. $x$ and $y$ are discrete real variables and discrete frequency variables, respectively. In this paper, the $z_0$ is represented as a 2D matrix and operated according to Eq. 22.

**FFHQ Model.** We adopt the ffhq_10m.pt with a size of 357.1MB as the baseline model for all the experiments relating to the FFHQ dataset.

### A.3 HYPERPARAMETER SENSITIVITY ANALYSIS

The hyperparameters used for PostEdit are listed in Tab.3, and its performance under different settings is presented in Tab.4. We conduct the following hyperparameters sensitivity analysis:

- **Appearance Probability of 0 in Matrix $P$.** A higher probability (*e.g.*, 0.7) improves **CLIP Similarity** but degrades **Background Preservation** metrics (*e.g.*, PSNR and SSIM). Conversely, a lower probability (*e.g.*, 0.3) favors background preservation at the expense of CLIP similarity.

- **Optimization Steps.** Reducing the number of steps, such as 50, decreases computation time but negatively impacts performance across most metrics. Increasing the steps to 150 offers slight performance improvement but reduces efficiency. The chosen configuration of 100 steps strikes a balance between quality and runtime.

- **Weighting Coefficient.** Setting $w$ to zero results in poor performance for both background preservation and editing capabilities. While increasing $w$ enhances background consistency, editing performance remains suboptimal.
- **Our Configuration.** The default settings strike a balance across all metrics, achieving competitive results in background preservation, editing alignment, and efficiency.

### A.4 Comparison The Images Layout of Different Datasets

In this section, we present a comparison of the layouts of the estimated $x_0$ at different intermediate timesteps, as inferred by the diffusion models trained on the SD and FFHQ datasets respectively.

In Fig. 6, we present three independent sets of results for both SD and FFHQ, each containing nine different instances of $\hat{z}_0$ selected from outputs of various iterations. The first three rows display the results for SD, while the remaining rows correspond to FFHQ. Each set is tasked with generating the same target image based on the same initial image. From left to right, the level of noise progressively decreases.

Notably, the layouts for SD are more varied, with inconsistencies in the cat's appearance, its position relative to the mirror, and the mirror's appearance across the three images. This contrasts sharply with the results from FFHQ, where the layouts consistently feature a centered face surrounded by a stable background.

To verify that this property is consistently observed in results based on the FFHQ model, we present additional examples in Fig. 7. As we move from bottom to top, the noise gradually decreases, while from left to right, there are 10 different examples. Each image represents the estimated $z_0$ from different iterations.

### A.5 Quantitative Results of Ablation Study

To better reflect the effectiveness of our adopted settings, we also conduct a quantitative results of the ablation study shown in Tab. 5. The results further verify the performance of all settings of PostEdit.

| Method | Background Preservation | | | | CLIP Similarity | | Efficiency |
| --- | --- | --- | --- | --- | --- | --- | --- |
| | PSNR$^\uparrow$ | LPIPS$^\downarrow_{\times 10^2}$ | MSE$^\downarrow_{\times 10^3}$ | SSIM$^\uparrow_{\times 10^2}$ | Whole$^\uparrow$ | Edited$^\uparrow$ | Time$^\downarrow$ |
| No Posterior Sampling | 21.31 | 16.88 | 10.36 | 73.21 | 26.38 | 23.28 | ∼**1s** |
| No mask | **28.31** | **4.61** | **2.64** | **84.15** | 25.17 | 20.94 | ∼1.5s |
| No $z_{in}$ | 24.01 | 10.92 | 6.24 | 77.28 | 26.45 | 23.45 | ∼1.5s |
| Ours Full | 27.04 | 6.38 | 3.24 | 82.20 | **26.76** | **24.14** | ∼1.5s |

Table 5: **Quantitative Comparisons of Ablation Study.**

### A.6 Results of Long-text Editing

We conducted long-text editing experiments, with the results presented in Fig. 8, Fig. 9, Fig. 10 and Fig. 11. These results demonstrate that the editing capabilities of PostEdit extend beyond simple word replacements.

### A.7 More Editing Results

Here, we qualitatively compare with other baselines including NPI, GR, DI, and IP2P, SeedX. The results are shown in Fig. 12. The results support the conclusion in the manuscript.

### A.8 Reconstruction Results of Different Forward Operators

Images in Fig. 14 are reconstructed through different forward operators as shown in Eq. 17 and Eq. 21. The corresponding quantitative comparison of image reconstruction of different measurements is shown in Tab. 6. The corresponding quantitative comparison of image reconstruction of different measurements is shown in Tab. 6.

| Measurement | Background Preservation | | | | Efficiency |
|---|---|---|---|---|---|
| | PSNR$^\uparrow$ | LPIPS$^\downarrow_{\times 10^2}$ | MSE$^\downarrow_{\times 10^3}$ | SSIM$^\uparrow_{\times 10^2}$ | Time$^\downarrow$ |
| Eq. 21 | 24.90 | 7.60 | 4.31 | 74.03 | $\sim$ 1.5s |
| Eq. 17 (Used) | 24.39 | 9.00 | 4.75 | 72.74 | $\sim$ 1.5s |

Table 6: **Quantitative Comparisons of Image Reconstruction using different Measurements.**

## A.9 INTERMEDIATE RESULTS

The intermediate state for different iterative steps are shown detailed in Fig. 13.

## A.10 ADDITIONAL RECONSTRUCTION RESULTS

The additional reconstruction results are exhibited in Fig. 15 and Fig. 16. Additionally, We compare reconstruction quality of different methods show in Fig. 19. The results reflect the effectiveness of PostEdit to reconstruct high frequency information.

## A.11 COMPARISON BETWEEN LCM-SD1.5 AND SDXL-TURBO

Fig. 17 illustrates the inference speed of LCM-SD1.5, which is utilized in our method, alongside SDXL-Turbo. The results indicate that TurboEdit Deutch et al. (2024) may not be faster than our method, despite its reliance on the advanced baseline model, SDXL-Turbo. All experiments were conducted on a single NVIDIA A100 GPU with 80GB of memory.

## A.12 GENERATING EDIT INSTRUCTION FOR COMPARING WITH INSTRUCTION-BASED IMAGE EDITING APPROACHES

We provide GPT4-o with the following instruction shown in Fig. 18 to convert the differences between the input prompt and the edited prompt into editing instructions suitable for IP2P and SeedX.

## A.13 USER STUDY

We invited 34 anonymous volunteers to rank the preferred results of image editing results. The results are evaluated by the quality of background preservation and features aligned with the given target prompt. The feedback is shown in Tab. 7 and Tab. 8 and the preference represent a vote of the participants. The results indicate that PostEdit outperforms the compared baselines and is the most popular approach for both image reconstruction and editing tasks.

| Method | Ours | iCD | DI | SPD | NTI | PnP | DDCM |
|---|---|---|---|---|---|---|---|
| Preference (Editing) | **81** | 22 | 8 | 23 | 5 | 13 | 12 |

| Method | TurboEdit | NPI | GR | IP2P | SeedX | OmniGen | |
|---|---|---|---|---|---|---|---|
| Preference (Editing) | 24 | 18 | 30 | 1 | 1 | 55 | |

Table 7: **User Study of Image Editing.**

| Method | Ours | NTI | NPI | DDCM | iCD |
|---|---|---|---|---|---|
| Preference (Reconstruction) | **101** | 66 | 56 | 1 | 11 |

Table 8: **User Study of Image Reconstruction.**

---

**Algorithm 2** : Posterior Sampling for Image Reconstruction

---

**Require:** Diffusion model $\epsilon_\theta$, diffusion sampler $\hat{z}_0(\cdot)$, posterior sampling steps $N$, step size $h$, image $x_0$, measurement $y$, weight $w$, initial prompt $c_{ini}$, encoder $\mathcal{E}$, decoder $\mathcal{D}$, noise schedule $\alpha(t)$, $\sigma(t)$, optimization steps $N_L$ and posterior sampler sequence $\{\tau_i\}_{i=0}^N$.

$z_{\tau_N} \sim \mathcal{N}(\mathbf{0}, \boldsymbol{I})$.

**for** $i = N$ to $0$ **do**

    $z_0 = \hat{z}_0(z_{\tau_i}, \tau_i, c_{ini})$

    **for** $j = 0$ to $N_L$ **do**

        $\epsilon \sim \mathcal{N}(\mathbf{0}, \boldsymbol{I})$.

        $z_0^{(j+1)} = (1-w) \cdot z_0^{(j)} + w \cdot z_{in} - h \cdot \nabla_{z_0^{(k)}} \left( \frac{\|z_0^{(j)} - z_0\|^2}{2\sigma_t^2} + \frac{\|\mathcal{A}(z_0^{(j)}) - y\|^2}{2m^2} \right) + \sqrt{2h}\epsilon$.

    **end for**

    Sample $z_{\tau_{i-1}} \sim \mathcal{N}\left( z_0^{(N_L)}, \sigma_{\tau_{i-1}}\boldsymbol{I} \right)$.

**end for**

$x_0 = \mathcal{D}(z_0)$

**Return** $x_0$

---

### A.14 CLASSIFIER FREE DIFFUSION GUIDANCE

According to CFG Ho & Salimans (2022), the generation process is governed by the conditional score, which can be derived as follows

$$
\begin{aligned}
\nabla_{\boldsymbol{x}_t} \log p(\boldsymbol{x}_t \mid \boldsymbol{c}) &= \nabla_{\boldsymbol{x}_t} \log \left( \frac{p(\boldsymbol{x}_t)\, p(c \mid \boldsymbol{x}_t)}{p(c)} \right) \\
&= \nabla_{\boldsymbol{x}_t} \log p(\boldsymbol{x}_t) + p(c \mid \boldsymbol{x}_t) \\
&\quad - \nabla_{\boldsymbol{x}_t} \log p(c) \\
&= \nabla_{\boldsymbol{x}_t} \log p(\boldsymbol{x}_t) + \nabla_{\boldsymbol{x}_t} \log p(c \mid \boldsymbol{x}_t).
\end{aligned}
\tag{23}
$$

And then the term $\nabla_{\boldsymbol{x}_t} \log p(c \mid \boldsymbol{x}_t)$ can be derived as

$$
\begin{aligned}
\nabla_{\mathbf{x}_t} \log p(c \mid \mathbf{x}_t) &= \nabla_{\mathbf{x}_t} \log p(\mathbf{x}_t \mid c) - \nabla_{\mathbf{x}_t} \log p(\mathbf{x}_t) \\
&= -\frac{1}{\sqrt{1 - \bar{\alpha}_t}} \left( \epsilon_\theta(\mathbf{x}_t, t, c) - \epsilon_\theta(\mathbf{x}_t, t) \right).
\end{aligned}
\tag{24}
$$

Substituting the above term into the gradients of classifier guidance, we can obtain

$$
\begin{aligned}
\bar{\epsilon}_\theta(\mathbf{x}_t, t, c) &= \epsilon_\theta(\mathbf{x}_t, t, c) - \sqrt{1 - \bar{\alpha}_t}\, w \nabla_{\mathbf{x}_t} \log p(c \mid \mathbf{x}_t) \\
&= \epsilon_\theta(\mathbf{x}_t, t, c) + w(\epsilon_\theta(\mathbf{x}_t, t, c) - \epsilon_\theta(\mathbf{x}_t, t)) \\
&= (w+1)\epsilon_\theta(\mathbf{x}_t, t, c) - w\epsilon_\theta(\mathbf{x}_t, t).
\end{aligned}
\tag{25}
$$

Clearly, there is an unconditional term (also known as the null-text term) that directly contributes to the bias in the estimation of $x_0$ when the DDIM inversion process is applied under Classifier-Free Guidance (CFG) conditions. To mitigate this influence, a tuning process is typically required to optimize the null-text term, ensuring high-quality reconstruction. Furthermore, to achieve a better alignment between the generated image and the text prompt, as well as to enhance image quality, it is often necessary to utilize a larger value of $w$. However, this can exacerbate cumulative errors, leading to significant deviations in the acquired latent representation.

### A.15 ALGORITHM FOR IMAGE RECONSTRUCTION

The overall process of image reconstruction by applying posterior sampling is shown specifically in Alg. 2.

The differences between the reconstruction and editing tasks is the ODE solvers applied to estimate $\hat{z}_0$, and the initial and target prompts remain the same with each other for image reconstruction. All the parameter Settings are shown specifically in App. A.2.

### A.16 PROOF OF PROPOSITION 2

According to Eq. 14, the distribution of $\mathbf{z}_{t-1}$ depends on $\mathbf{z}_t$ and $\mathbf{z}_0$. The marginal distribution relating to timestep $t-1$ can be rewritten by Proof. We first factorize the measurement conditioned time-marginal $p\left(\mathbf{z}_{t_2} \mid \mathbf{y}\right)$ by

$$
\begin{aligned}
p\left(\mathbf{z}_{t-1} \mid \mathbf{y}, c\right) &= \iint p\left(\mathbf{z}_{t-1}, \mathbf{z}_0^w, \mathbf{z}_t \mid \mathbf{y}\right) \mathrm{d}\mathbf{z}_0^w \, \mathrm{d}\mathbf{z}_t \\
&= \iint p\left(\mathbf{z}_t \mid \mathbf{y}, c\right) p\left(\mathbf{z}_0^w \mid \mathbf{z}_t, \mathbf{y}, c\right) p\left(\mathbf{z}_{t-1} \mid \mathbf{z}_0^w, \mathbf{z}_t, y, c\right) \mathrm{d}\mathbf{z}_0^w \, \mathrm{d}\mathbf{z}_t,
\end{aligned}
\tag{26}
$$

according to the proposition 1, the above equation can be written as

$$
\begin{aligned}
p\left(\mathbf{z}_{t-1} \mid \mathbf{y}, c\right) &= \iint p\left(\mathbf{z}_t \mid \mathbf{y}, c\right) p\left(\mathbf{z}_0^w \mid \mathbf{z}_t, \mathbf{y}, c\right) p\left(\mathbf{z}_{t-1} \mid \mathbf{z}_0^w, \mathbf{z}_t, \mathbf{y}, c\right) \mathrm{d}\mathbf{z}_0^w \, \mathrm{d}\mathbf{z}_t \\
&= \iint p\left(\mathbf{z}_t \mid \mathbf{y}, c\right) p\left(\left((1-w) \cdot \mathbf{z}_0 + w \cdot \mathbf{z}_{in}\right) \mid \mathbf{z}_t, \mathbf{y}, c\right) \\
&\qquad p\left(\mathbf{z}_{t-1} \mid \left((1-w) \cdot \mathbf{z}_0 + w \cdot \mathbf{z}_{in}\right), \mathbf{z}_t, y, c\right) \mathrm{d}\left((1-w) \cdot \mathbf{z}_0 + w \cdot \mathbf{z}_{in}\right) \, \mathrm{d}\mathbf{z}_t \\
&\overset{\text{(i)}}{=} \iint p\left(\mathbf{z}_t \mid \mathbf{y}\right)\left[(1-w) \cdot p\left(\mathbf{z}_0 \mid \mathbf{z}_t, \mathbf{y}, c\right) + w \cdot p\left(\mathbf{z}_{in} \mid \mathbf{z}_t, \mathbf{y}\right)\right] \\
&\qquad p\left(\mathbf{z}_{t-1} \mid (1-w) \cdot \mathbf{z}_0, \mathbf{z}_t, y\right) \mathrm{d}\left((1-w) \cdot \mathbf{z}_0 + w \cdot \mathbf{z}_{in}\right) \, \mathrm{d}\mathbf{z}_t \\
&\overset{\text{(i)}}{=} \iint p\left(\mathbf{z}_t \mid \mathbf{y}\right) p\left((1-w) \cdot \mathbf{z}_0 \mid \mathbf{z}_t, \mathbf{y}, c\right) \\
&\qquad p\left(\mathbf{z}_{t-1} \mid (1-w) \cdot \mathbf{z}_0, \mathbf{z}_t, y\right) \mathrm{d}\left((1-w) \cdot \mathbf{z}_0\right) \, \mathrm{d}\mathbf{z}_t \\
&\overset{\text{(ii)}}{=} \iint p\left(\mathbf{z}_t \mid \mathbf{y}\right) p\left(\mathbf{z}_0 \mid \mathbf{z}_t, \mathbf{y}, c\right) p\left(\mathbf{z}_{t-1} \mid \mathbf{z}_0, \mathbf{z}_t, y\right) \mathrm{d}\mathbf{z}_0 \, \mathrm{d}\mathbf{z}_t \\
&= \iint p\left(\mathbf{z}_t \mid \mathbf{y}\right) p\left(\mathbf{z}_0 \mid \mathbf{z}_t, \mathbf{y}, c\right) p\left(\mathbf{z}_{t-1} \mid \mathbf{z}_0\right) \mathrm{d}\mathbf{z}_0 \, \mathrm{d}\mathbf{z}_t \\
&= \mathbb{E}_{\mathbf{z}_t \sim p(\mathbf{z}_t \mid \mathbf{y})} \mathbb{E}_{\mathbf{z}_0 \sim p\left(\mathbf{z}_0 \mid \mathbf{z}_{t_1}, \mathbf{y}, c\right)} p\left(\mathbf{z}_{t-1} \mid \mathbf{z}_0\right) \\
&\overset{\text{(iii)}}{=} \mathbb{E}_{\mathbf{z}_0 \sim p(\mathbf{z}_0 \mid \mathbf{z}_t, \mathbf{y}, c)} \mathcal{N}\left(\mathbf{z}_{t-1}; \mathbf{z}_0, \sigma_{t-1}^2 \boldsymbol{I}\right),
\end{aligned}
\tag{27}
$$

where (i) is dues to independent relationships and (ii) is derived by variable substitution and $c$ is the given target prompt. (iii) is derived directly according to the process defined in Eq. 15, whose independent variant is substituted by $\mathbf{z}_0$ instead of $\mathbf{z}_0^w$.

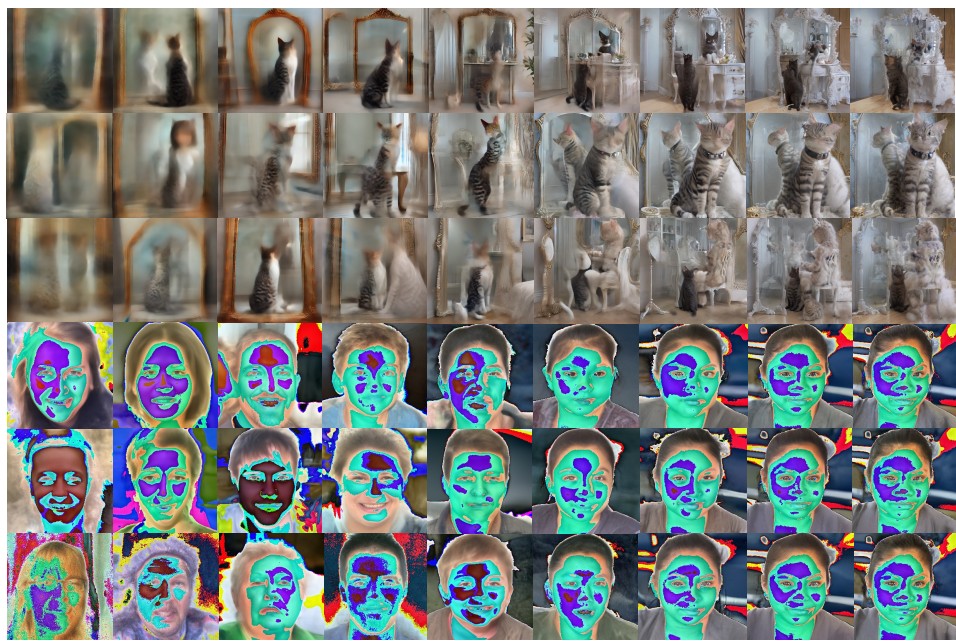

Figure 6: Layouts of evaluated outputs for the same objects at different intermediate timesteps.

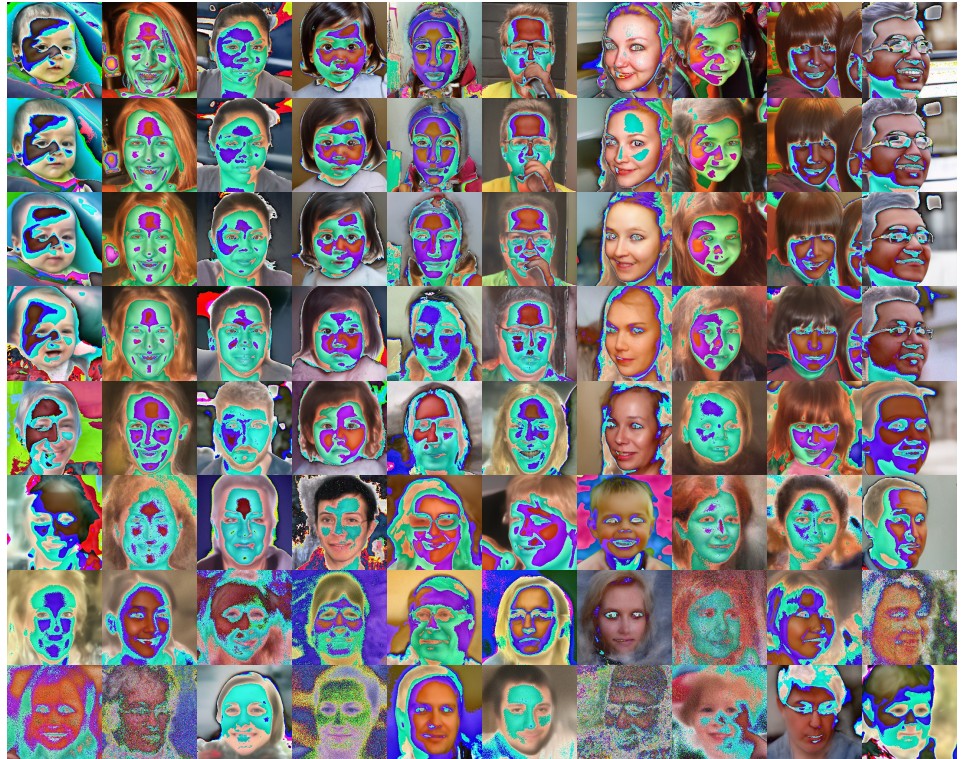

Figure 7: Layouts of evaluated output for various objects and timesteps.

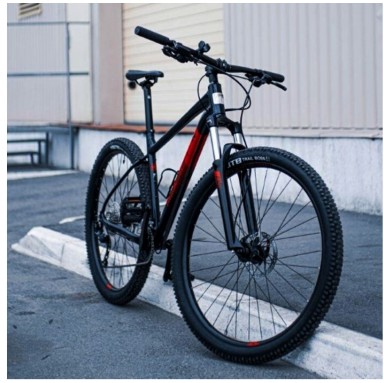 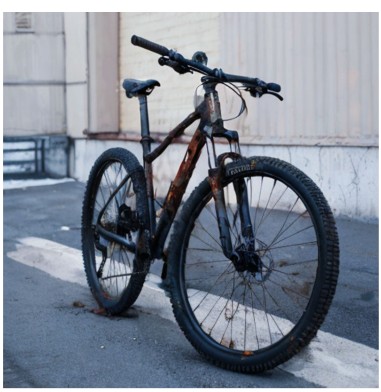

Original (Left):
A sleek mountain bicycle is parked on a slight slant, positioned on the edge of a paved road near a modern building with beige walls. The bike features a black and red frame, with thick treaded tires designed for rugged terrain. The handlebars are straight with ergonomic grips, and the front suspension fork indicates it's built for off-road performance. The surrounding area is industrial, with concrete curbs, metallic railings, and muted urban tones adding to the setting's atmosphere. The lighting is natural, softly highlighting the bicycle's glossy finish and sturdy construction.

Edited (Right):
A completely rusty mountain bicycle is parked on a slight slant, positioned on the edge of a paved road near a modern building with beige walls. The entire frame, including the handlebars, suspension fork, and wheel rims, is heavily corroded, covered in a thick layer of rust that has overtaken any remnants of paint or gloss. The chain and gear system are seized and crusted with rust, rendering them non-functional. The tires are cracked and worn, with rust even creeping onto the spokes and hubs. The industrial setting, with concrete curbs and metallic railings, provides a stark contrast to the bike's decayed and abandoned appearance. The natural lighting accentuates the uneven texture and reddish-brown hue of the rust, making the bicycle look as if it has been left to the elements for many years.

Figure 8: **Example of Long-text Editing.**

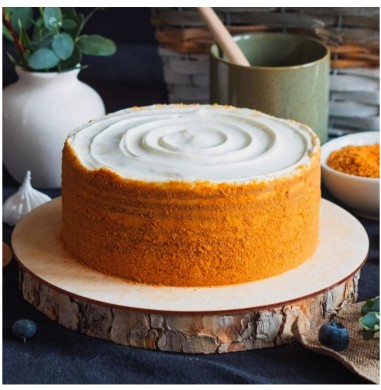 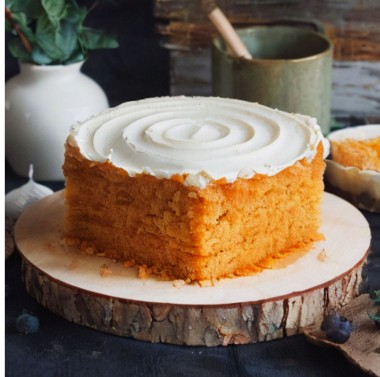

Original (Left):
A beautifully crafted round cake with a rustic charm sits elegantly on a wooden plate made from a natural tree slice, showcasing its earthy aesthetic. The cake is adorned with a light orange frosting, finely textured with a crumb coating that wraps around its sides, giving it a warm and inviting appearance. The top of the cake features a smooth swirl of white cream, adding a touch of contrast and sophistication to the design. Surrounding the cake is a cozy scene, featuring elements like a white ceramic vase with fresh green leaves, a small bowl filled with vibrant orange crumbs, and a muted green mug with a wooden spoon resting nearby. The setting is completed with a dark fabric background and a woven basket, creating a homey and natural ambiance that complements the rustic presentation of the cake.

Edited (Right):
A beautifully crafted square cake with a rustic charm sits elegantly on a wooden plate made from a natural tree slice, showcasing its earthy aesthetic. The cake is adorned with a light orange frosting, finely textured with a crumb coating that wraps around its sides, giving it a warm and inviting appearance. The top of the cake features a smooth swirl of white cream, adding a touch of contrast and sophistication to the design despite its square shape. Surrounding the cake is a cozy scene, featuring elements like a white ceramic vase with fresh green leaves, a small bowl filled with vibrant orange crumbs, and a muted green mug with a wooden spoon resting nearby. The setting is completed with a dark fabric background and a woven basket, creating a homey and natural ambiance that complements the rustic presentation of the square cake.

Figure 9: **Example of Long-text Editing.**

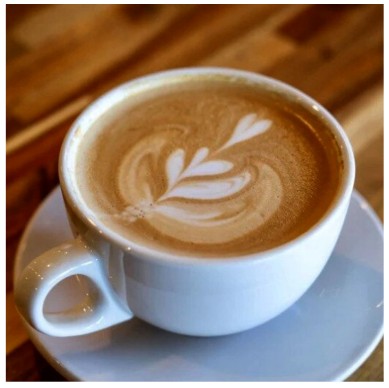 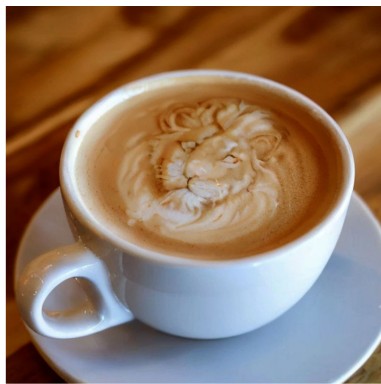

Original (Left):
A white ceramic cup filled with a beautifully crafted latte sits on a matching white saucer, placed on a polished wooden table. The latte art on the surface features an intricate leaf-like design made with steamed milk, showcasing the skill of the barista. The warm tones of the coffee complement the rich wood grain of the table, creating a cozy and inviting atmosphere.

Edited (Right):
A white ceramic cup filled with a beautifully crafted latte sits on a matching white saucer, placed on a polished wooden table. The latte art on the surface has been transformed into the detailed outline of a lion's head, crafted with precision using steamed milk. The lion's mane and facial features are depicted with smooth and intricate swirls, showcasing an impressive artistic skill. The warm tones of the coffee remain unchanged, blending harmoniously with the rich wood grain of the table, preserving the cozy and inviting atmosphere.

Figure 10: **Example of Long-text Editing.**

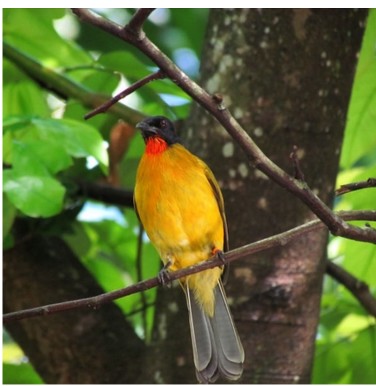 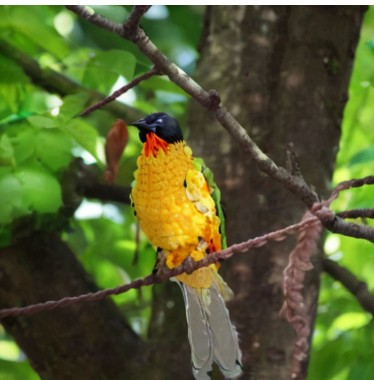

Original (Left):
A vibrant bird with bright yellow feathers and an orange-red chest perches gracefully on a thin tree branch in a lush, green forest. The bird's sharp black head contrasts beautifully with its colorful plumage. The background is filled with soft-focus green leaves and the textured bark of trees, creating a serene and natural setting.

Edited (Right):
A fully crocheted bird, crafted entirely from colorful yarn, sits delicately on a thin tree branch in a lush, green forest. The bird's body is made of soft, bright yellow yarn, with an orange-red chest, and its head is crocheted in black with visible loops and stitches, giving it a distinctly handmade appearance. The wings, tail, and legs are intricately shaped with consistent crochet patterns, making the bird look entirely crafted from yarn. The artificial, textured crochet design stands out against the natural elements, with the tree branch and the vibrant green leaves in the background remaining unchanged. The setting maintains its serene, natural atmosphere, while the fully crocheted bird adds a whimsical and creative touch.

Figure 11: **Example of Long-text Editing.**

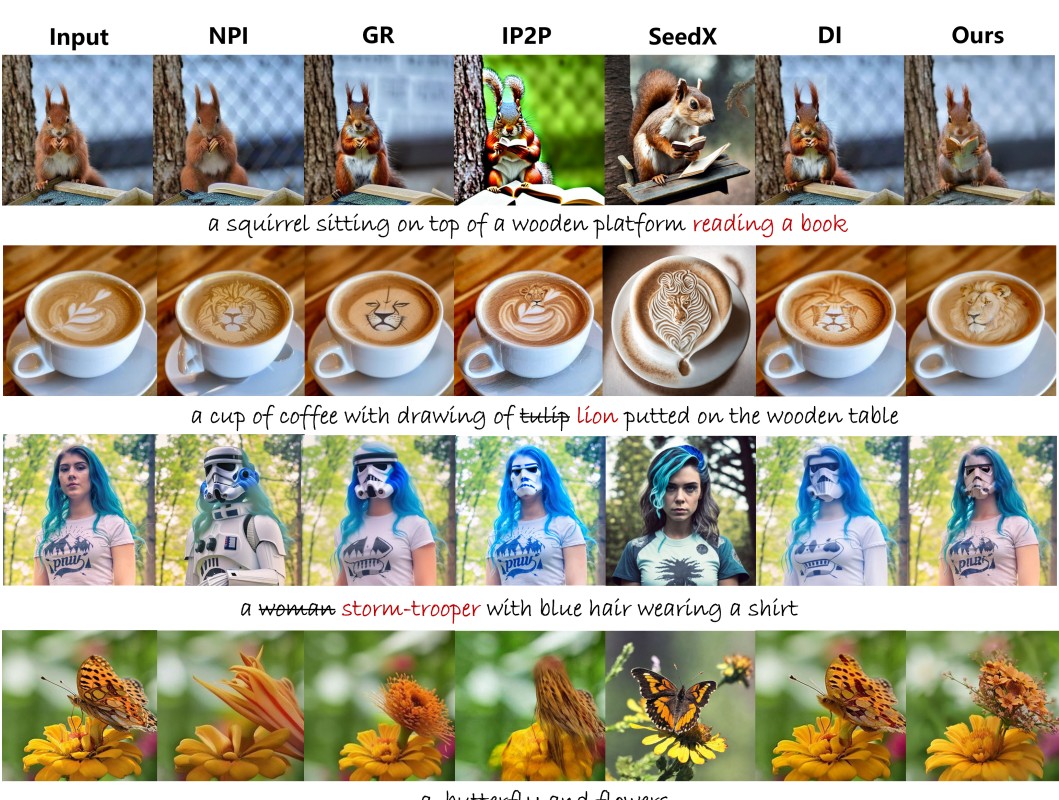

Figure 12: **Additional Comparison Results of The Remained Baselines.**

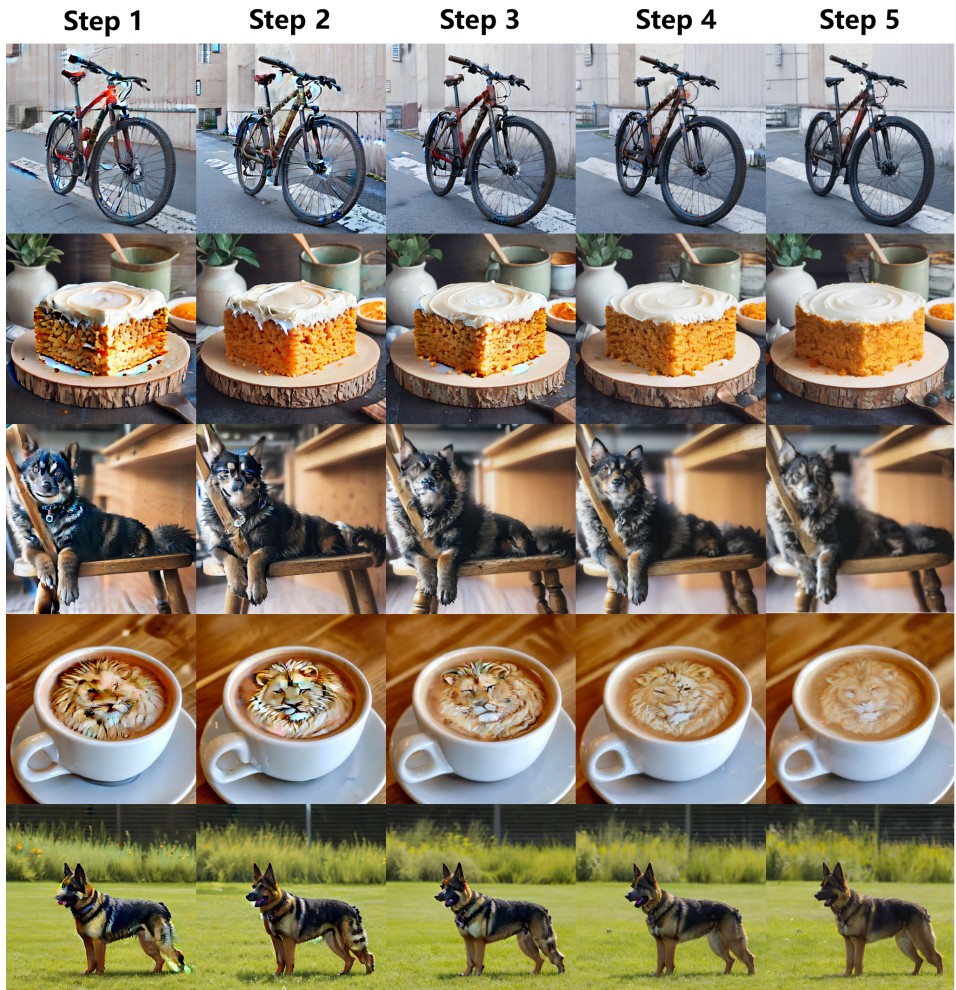

Figure 13: **Intermediate Results.**

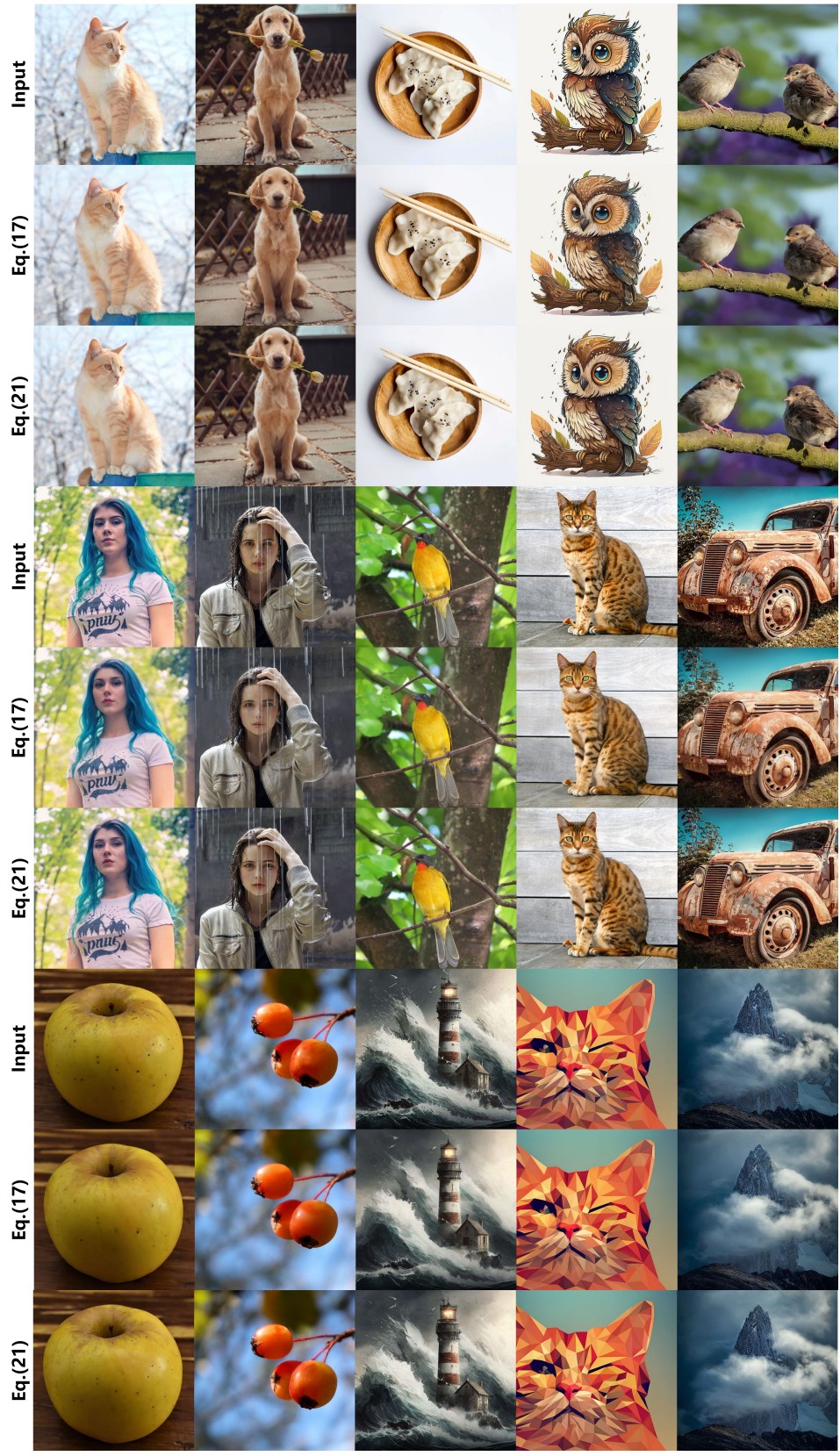

Figure 14: **Additional Comparison Results Based on Different Measurements.**

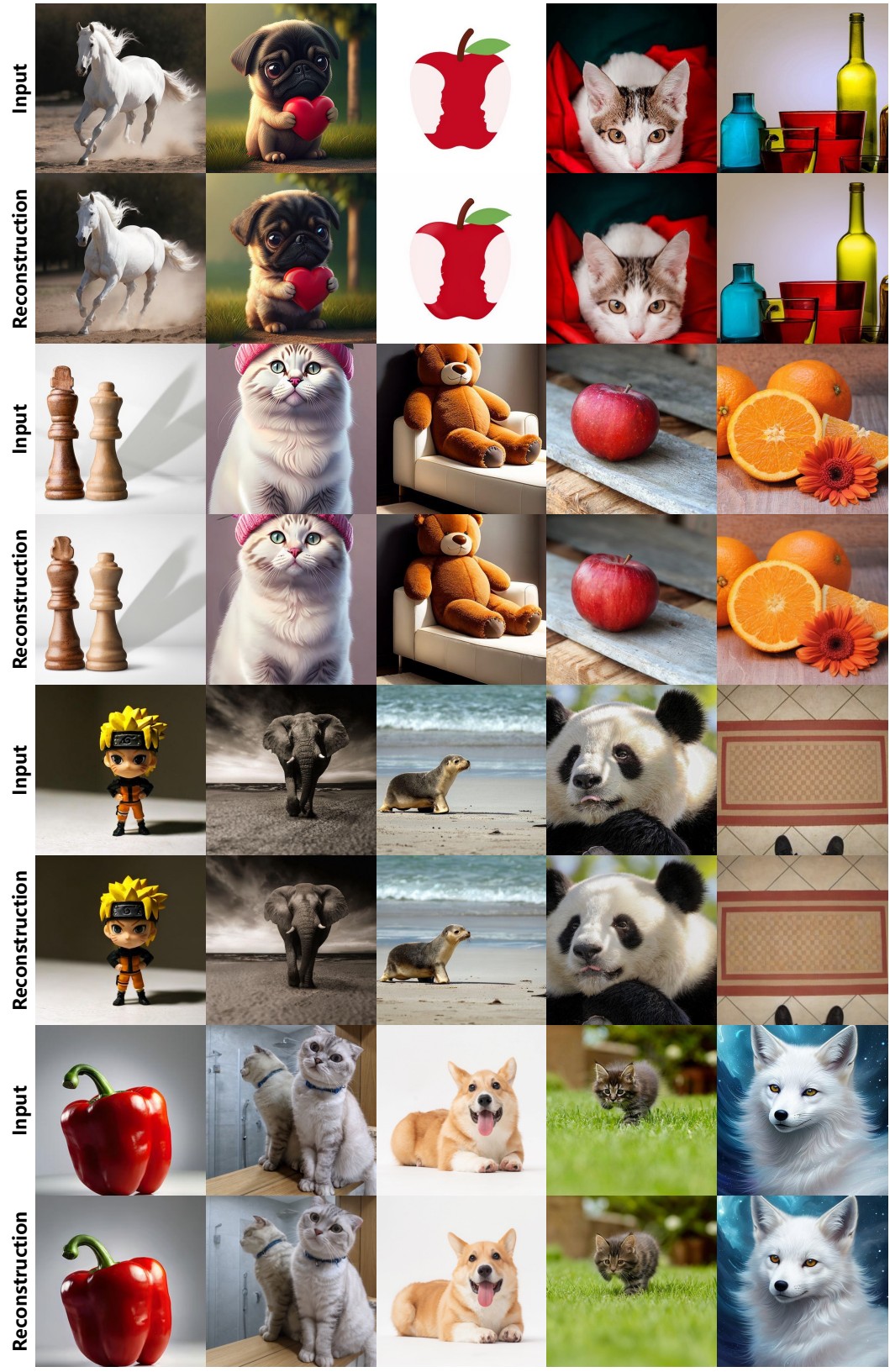

Figure 15: **Additional reconstruction Results.**

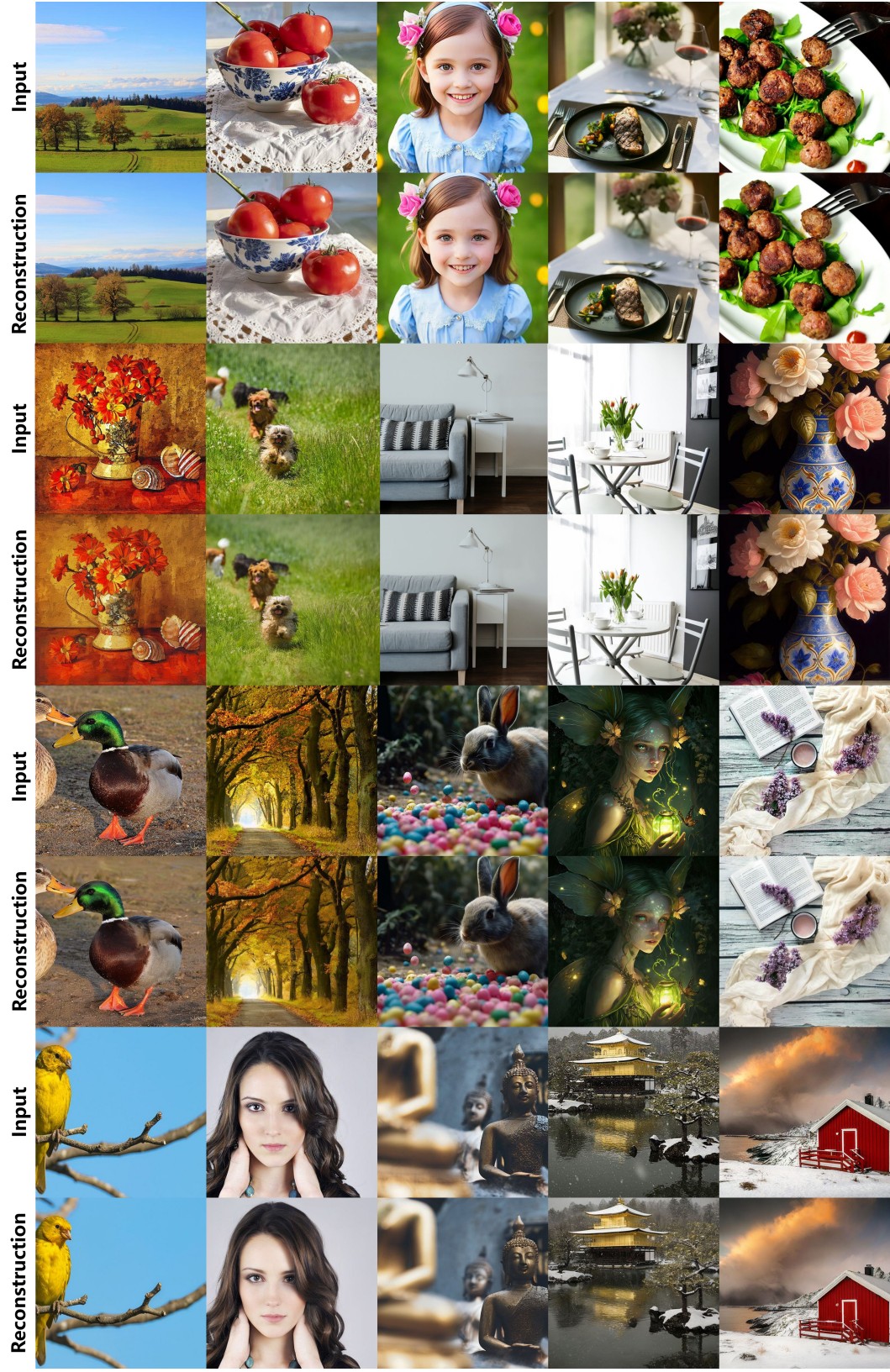

Figure 16: **Additional reconstruction Results.**

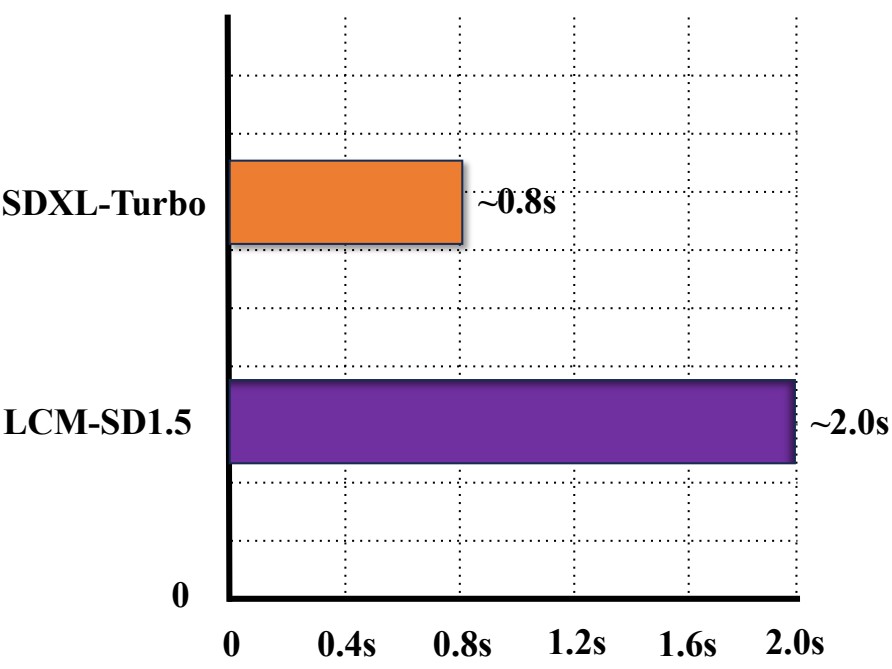

Figure 17: **Comparison of Inference Speed.**

You are an expert in image editing tasks, responsible for generating clear and concise editing instructions based on two image descriptions.

Given the following descriptions:

1. Original image description: "{original_prompt}"

2. Edited image description: "{edited_prompt}"

Please infer the differences between these descriptions and provide a single concise instruction to describe the necessary edits to the original image. The instruction should be as brief as possible. For example:

- If the difference is a color change, use expressions like "Make the bicycle rusty."

- If it involves adding elements, use expressions like "Add a red hat to the person."

- If it involves removing content, use expressions like "Remove the text from the image."

Output format:

```json
{
    "instruction": "Your instruction content"
}
```

Figure 18: **Instruction Generation process GPT.**

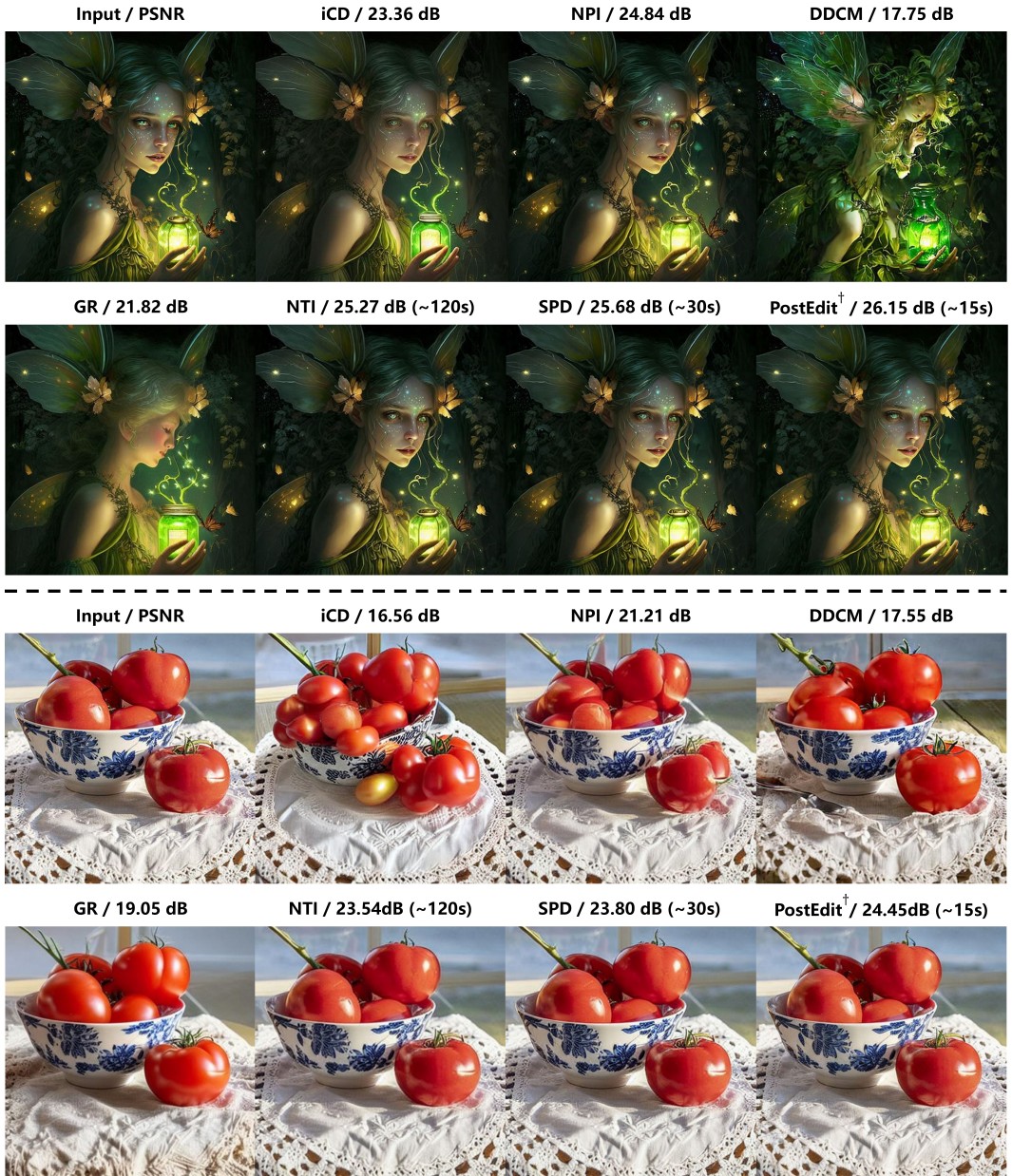

Figure 19: **Comparison of Different methods for Reconstruction of High Frequency Details.** †
represents 10000 optimization steps are adopted for this result.

