# OpenReview forum: "PostEdit: Posterior Sampling for Efficient Zero-Shot Image Editing"
_ICLR.cc/2025/Conference — ICLR 2025 Poster_

### Official Review · Reviewer_CdSe · 2024-11-02

**Soundness:** 3
**Presentation:** 3
**Contribution:** 3
**Rating:** 6
**Confidence:** 5

**Summary:**

This paper introduces PostEdit, a method that incorporates a posterior scheme to govern the diffusion sampling process. Specifically, a corresponding measurement term related to both the initial features and Langevin dynamics is introduced to optimize the estimated image generated by the given target prompt.

**Strengths:**

This paper introduces PostEdit, a method that incorporates a posterior scheme to govern the diffusion sampling process. Specifically, a corresponding measurement term related to both the initial features and Langevin dynamics is introduced to optimize the estimated image generated by the given target prompt.

**Weaknesses:**

1. The reviewer encountered several points in the algorithm that were difficult to understand, raising the possibility of inaccuracies. The reviewer requests that the authors provide clear responses to the following questions: (1) In line 4, does \(z_0\) bear no relation to the original image, given that \(z_N\) is derived from pure noise? (2) Can lines 6 to 9 be interpreted as a re-generation of \(z_0\)? (3) Is the alignment with the original image (i.e., \(z_{in}\)) genuinely implemented only in lines 12 to 14? (4) In line 15, the reviewer finds the superscript \(N\) on \(z^{(N)}\) unusual, as it would conventionally be \(m\) in this context. Additionally, the reviewer is puzzled by the disappearance of \(z_{\tau_{i-1}}\) in the next loop iteration. These uncertainties make the methodology difficult to follow, and the reviewer suggests clarifying these points for future readers.

2. Regarding the essence of the proposed method, the reviewer interprets it as generating an image that aligns with the target prompt, followed by posterior probability correction techniques that reintroduce information from the original image via weighted blending, masking, or similar means. The reviewer has experience with studies on posterior probability and would appreciate confirmation.

3. The reviewer finds the reported speed and runtime of the proposed method particularly perplexing. Given the numerous loops in the algorithm, it is unclear how the method achieves a runtime of only 1.2 seconds. As far as the reviewer is aware, most training-free image editing methods, even with fewer layers in the algorithmic structure, typically take tens of seconds. Could the authors clarify where this dramatic time saving is achieved? Beyond seeing a 1.2-second result, the reviewer would like a breakdown of time components comparing this method to other training-free methods.

4. The number of baseline methods in the comparative experiments is insufficient, with few comparisons to state-of-the-art training-based methods. This lack of comparison feels inequitable, as training-based methods have become well-established for image editing, showing robustness and speed (e.g., InstructP2P, SmartEdit, SEED-X-Edit). The reviewer questions what specific contributions are offered by the proposed training-free method.

5. The proposed method involves an overwhelming number of hyperparameters; there appear to be countless hyperparameters even within the algorithm section alone. The reviewer believes that a detailed hyperparameter sensitivity analysis is necessary to ensure that the model's robustness is not significantly affected by the choice of hyperparameters. From the reviewer's experience, many training-free methods require intricate hyperparameter tuning to achieve optimal results. Therefore, the reviewer requests that the authors provide the specific hyperparameters used for each image in the experimental section.

6. The related work section appears to reference too few relevant studies in the image editing field, which is rapidly evolving. The reviewer suggests expanding this section by referencing additional survey papers to strengthen the literature review.

**Questions:**

See Weaknesses.

---

> ### Author Response · Authors · 2024-11-22
> **Reply for Reviewer CdSe**
>
> **Thank you for your valuable review and suggestions. Below, we address each comment point-by-point.  We are looking forward to your further discussion and questions**
>
> > **W1.** Further clarification on the algorithm.
>
> > (1) In line 4, does ($z_0$) bear no relation to the original image, given that ($z_N$) is derived from pure noise?
>
> Yes. $z_0$ is sampled from pure noise using the LCM sampler conditioned on the target prompt, ensuring no direct relation to the original image. As detailed in the scheduler settings in **Appendix A.2**, this approach significantly improves efficiency without compromising generation quality. While $z_0$ has minimal relation to the original image, PostEdit enhances both editing and reconstruction capabilities through the subsequent optimization process.
>
> > (2) Can lines 6 to 9 be interpreted as a re-generation of ($z_0$)?
>
> Yes. Lines 6–9 of **Algorithm 1** describe our multi-step solver for sampling $z_0$ from $z_t$. This multi-step strategy enhances the quality of generation, ensuring alignment with the target prompt.
>
> > (3) Is the alignment with the original image (i.e., ($z_{in}$)) genuinely implemented only in lines 12 to 14?
>
> Yes. Lines 12–14 incorporate $z_{in}$ to address the trade-offs between reconstruction and editing, as well as the complexities of large-scale text-to-image models. This process is illustrated in **Figures 6 and 7** of **Appendix A.4**. The value of $w$ is set to 0.1, with detailed parameter settings provided in the "Hyper-parameters in Alg. 1" section of **Appendix A.2**.
>
> > (4) In line 15, the reviewer finds the superscript (N) on (z^{(N)}) unusual, as it would conventionally be (m) in this context. Additionally, the reviewer is puzzled by the disappearance of (z_{\tau_{i-1}}) in the next loop iteration.
>
> Sincerely thank you for pointing out this mistake, we have seriously revised the Algorithm 1. Below, we outline the key steps for clarification:
> - Line 2: Encode an initial image using the encoder and let $z_{in}=z_0$ to save the current $z_0$ since it is updated in the following steps.
> - Line 5: Random noise is initially added to $z_0$ following the DDPM scheduler.
> - Line 6: A multi-step diffusion sampler is utilized to generate the updated $z_0$, conditioned on the specified target prompt.
> - Line 8: Assign an initial value to the variable $z_0^{(k)}$ in the next loop.
> - Line 10: Random noise is sampled at each iterative step to facilitate Langevin dynamics. To ensure variability, the random noise must differ across rounds.
> - Line 11: Our optimization process is applied to refine $z_0$ by leveraging two gradient terms: one aligns $z_0$ with the target prompt, and the other integrates background features. To ensure the global optimal solution, random noise is incorporated, introducing Langevin dynamics. The specific values of all parameters used in the equation are detailed in Appendix A.2.
> - Line 13: Random noise is added to the optimized $z_0$ to obtain $z_{\tau_{i-1}}$ , enabling sampling in the direction of the clear image. The time sequences and \{$\tau_i$\} and \{$t_j$\} are both subsets of the time sequence defined by the DDPM scheduler. Although the number of steps differs between the two sequences, some time steps overlap in certain iterations. For instance, $\tau_{i}=t_{n-1}$.
>
> > **W2.** Further confirmation on the essence of the proposed method .
>
> Yes, the posterior probability plays a central role in our approach. We have developed a novel sampling process specifically designed to tackle the challenges associated with large-scale text-to-image reconstruction and editing tasks. In comparison to existing fine-tuning or training-based methods, our approach is more resource-efficient and demonstrates greater generalizability.
>
> > **W3.** More clarifications on the runtime.
>
> > (1) Given the numerous loops in the algorithm, it is unclear how the method achieves a runtime of only 1.2 seconds. Could the authors clarify where this dramatic time saving is achieved?
>
> The significant time saving in our method is achieved through the following key factors:
>
> 1. **Noise Sampling**: The added noise is directly sampled from Gaussian noise, without requiring inference from a network (e.g., the Unet of Stable Diffusion).
> 2. **Network-Independent Optimization**: The optimization process is independent of any network, enabling the measurement $\boldsymbol{y}$ and $z_0$ to be computed directly without additional computational overhead.
> 3. **Efficient Sampling Steps**: PostEdit employs the highly efficient LCM sampler, which significantly reduces the number of sampling steps. Specifically, to achieve both high efficiency and superior generation quality, we perform only 5 denoising steps to produce a clean image.

---

> ### Author Response · Authors · 2024-11-22
> **Reply for Reviewer CdSe 2**
>
> > (2) Beyond seeing a 1.2-second result, the reviewer would like a breakdown of time components comparing this method to other training-free methods.
>
> The detailed time component analysis is shown in the table below:
>
> | **Method**   | **Inversion Process (s)**     | **Generation Process (s)** | **Optimization Process (s)** |
> |--------------|-------------------------------|-----------------------------|------------------------------|
> | NTI          | 110 (500 steps null-text optimization) | 10 (50 steps DDIM using SD1.5) | Mixed with inversion         |
> | NPI          | 5 (50 steps negative prompt inversion) | 10 (50 steps DDIM using SD1.5) | \                            |
> | iCD          | 0.8 (4 steps with reverse consistency) | 0.8 (4 steps forward consistency) | \                            |
> | DDCM         | \                             | 2 (12 steps with LCM UNet)  | \                            |
> | TurboEdit    | \                             | 1.2 (4 steps with SDXL-Turbo) | \                            |
> | **Ours**     | \                             | 0.8 (5 steps with LCM UNet) | 0.7 (100 steps)              |
>
> ### Explanation:
> - **Inversion Process**: Our method does not require an explicit inversion process, which eliminates a significant computational bottleneck present in methods like NTI or NPI.
> - **Generation Process**: By employing only 5 steps with the LCM UNet, we reduce the generation time to 0.8 seconds without compromising quality.
> - **Optimization Process**: The optimization process (e.g., refining $z_0$) involves only 100 steps and is completed within 0.7 seconds, contributing to the overall efficiency.
>
> Our method demonstrates superior speed compared to other training-free methods, while maintaining high editing and reconstruction quality, as illustrated in Figure 2 ,15 and 16.
>
> > **W4.** Add comparisons to training-based baselines, such as InstructP2P, SmartEdit, SEED-X-Edit.
>
> Thank you for the suggestion! We have supplemented comparisons to InstructP2P, SEED-X-Edit, and OmniGen (a recently open-sourced state-of-the-art approach). The quantitative and qualitative results are reported in **Table 2, Figure 4 and 12**. Since these methods require editing instructions (e.g., "Add clouds to the sky"), we use GPT-4o to generate editing instructions based on the source and target prompts.
>
> From the results, the advantages of training-free methods such as PostEdit can be summarized as follows:
> 1. **Generalization Ability**: Unlike training-based methods that require modifying and fine-tuning conditional U-Nets (e.g., InstructP2P expands the input channel of the U-Net from 4 to 8), PostEdit works directly with the original text-to-image generation models without any architectural changes or fine-tuning. This allows PostEdit to fully retain the exceptional generative capabilities of the base models, ensuring superior generalization across diverse editing tasks.
> 2. **Resource Efficiency**: Training-based methods require millions of paired training examples and significant computational resources, such as tens of modern GPUs, to achieve good performance. Additionally, obtaining high-quality paired data (input image + edited image + editing instruction) is highly challenging. In contrast, PostEdit eliminates the need for such costly resources.
> 3. **Flexibility**: PostEdit is model-agnostic, allowing it to be applied to any text-to-image diffusion model without retraining. Training-based methods, on the other hand, require retraining for newly released models, limiting their adaptability.
>
> These advantages highlight the robustness and practicality of PostEdit compared to training-based approaches.
>
> > **W5.** Adding a detailed hyperparameter sensitivity analysis to evaluate the robustness of the proposed algorithm, and provide the specific hyperparameters used for each image in the experimental section.
>
> In our experiments, all images are generated using the same hyper-parameter setting, described in **Appendix A.2**, to ensure consistency and robustness without specific tuning for individual cases.
>
> The hyperparameters of our algorithm include:
> - **f**: Appearance probability of 0 in the mask matrix,
> - **w**: Weighting coefficient introduced in **Proposition 1**,
> - **Optimization Steps**: The number of steps used in the optimization process.
>
> A comprehensive introduction to all hyperparameters is provided in **Appendix A.2** The results of the hyperparameter sensitivity analysis are presented in **Table 3 and 4 of Appendix A.3**, demonstrating the robustness of our method to variations in hyperparameter choices.
>
> > **W6.** Referencing additional survey papers to strengthen the literature review.
>
> We have added the related work section in Appendix A.1. Please check it.

---

> > ### Comment · Reviewer_CdSe · 2024-11-26
> > **Reviewer's feedback**
> >
> > Thanks for the clarification the authors provided and the efforts they made. After reading the response and other reviewers' comments, I maintain my rating.

---

> > > ### Author Response · Authors · 2024-11-27
> > > **Reply for Reviewer CdSe 3**
> > >
> > > Sincerely thank you for your dedication and professional comments.

---

### Official Review · Reviewer_98Bk · 2024-11-03

**Soundness:** 3
**Presentation:** 3
**Contribution:** 3
**Rating:** 6
**Confidence:** 4

**Summary:**

The paper introduces "PostEdit," a novel inversion- and training-free approach leveraging posterior sampling for efficient zero-shot image editing. The method addresses significant challenges in image editing, including controllability, background preservation, and efficiency. The overall results indicate promising performance, particularly in maintaining high-quality background similarity and efficiency in execution time. Nevertheless, there are a few methodological and experimental aspects that require further clarification and expansion.

**Strengths:**

1. The proposed method's efficiency in terms of GPU memory and inference time is noteworthy. Achieving high-quality results in approximately 1.5 seconds and with only 18 GB of GPU memory is an impressive step forward for zero-shot image editing.
2. The authors provide strong mathematical support for their approach, with the incorporation of Langevin dynamics and a posterior measurement term to optimize the estimated image. This theoretically addresses the issue of error accumulation seen in existing approaches.
3. The results—both qualitative and quantitative—demonstrate the approach's effectiveness, showing its superiority compared to some existing methods.

**Weaknesses:**

1. Methodological Clarifications:

a. Figure 2: Step 3 in Figure 2 is challenging to understand and requires more explanation regarding the optimization process. It would be beneficial to clearly explain how Langevin dynamics and the measurement term are incorporated in this step to improve the optimization.

b. The explanation of Symbols such as $\mathcal{A}$ are missing: In Equation (16), the symbol $\mathcal{A}$ is not clearly defined or explained. Adding a clear definition for this symbol would enhance the reader's understanding. Eq.16 needs more explanations for specific symbols such as m, h, and etc.

c. In lines 321-322, the authors mention the mask, however, how to obtain the mask, it seems that the paper without further explanations.

2. Experimental Clarifications:

a. Baselines: The experimental comparisons do not include state-of-the-art methods such as SPD Inversion [1], and Guide-and-Rescale [2]. Including these baselines is crucial to substantiate the claims of superiority.

b. Ablation Study: The ablation study lacks quantitative results to further support the analysis. Numerical data on the effectiveness of each component (e.g., posterior sampling vs. inversion process) would add robustness to the conclusions.

c. User Study: Given that image editing quality can be somewhat subjective, a user study would be helpful to establish the practical effectiveness of the proposed method more convincingly.

d. Complex Images: All the reconstruction results shown in the paper involve images with a single object and simple layouts. However, some inversion methods struggle with reconstructing images containing multiple detailed objects. Could you provide reconstruction results for such complex images to strengthen your claims about the method's versatility?


References for Missing Baselines:
[1] Li R B, Li R H, Guo S, Zhang L. Source Prompt Disentangled Inversion for Boosting Image Editability with Diffusion Models. European Conference on Computer Vision. 2024.

[2] Titov V, Khalmatova M, Ivanova A, et al. Guide-and-Rescale: Self-Guidance Mechanism for Effective Tuning-Free Real Image Editing. European Conference on Computer Vision. 2024.

**Questions:**

Please refer to the "Weaknesses" section above for my questions and concerns.

---

> ### Author Response · Authors · 2024-11-22
> **Reply for Reviewer 98Bk**
>
> **Thank you for your valuable review and suggestions. Below, we address each comment point-by-point. **We are looking forward to your further discussion and questions.****
>
> > **W1(a)**. More explanation regarding the optimization process. How Langevin dynamics and the measurement term are incorporated in this step to improve the optimization.
>
> **The Optimization Process** is aimed to make $\boldsymbol{z}_0$ align better with target prompt while maintaining background preservation. Here are more details explanation of our algorithm for better understanding: The optimization process (Step 3 in Fig. 2) and how Langevin dynamic is integrated are formulated in Eq. (16). The optimization involves three terms:
>
> - $\nabla_{\boldsymbol{z}_{0}^{(k)}}\left(\frac{\Vert z_0^{(k)}-z_0\Vert^2}{2\sigma_t^2}\right)$ for alignment with features of target prompt.
> - $\nabla_{\boldsymbol{z}_{0}^{(k)}}\left( \frac{\Vert\mathcal{A}\left(\boldsymbol{z}_0^{(k)}\right)-\boldsymbol{y}\Vert^2}{2m^2}\right)$ for background preservation.
> - $\sqrt{2h}\epsilon$ represents Langevin dynamics for global optima.
>
> The first two are used to perform gradient descent on the current timestep $z_0^{k}$ to improve editing accuracy and background preservation. The last one is applied to introduce perturbation to avoid local optimal solutions.
>
> **Effects of The Measurement Term $\boldsymbol{y}$.** The measurement  $\boldsymbol{y}$ is designed to encapsulate partial information from both the initial image, achieved through element-wise multiplication with a mask matrix. Given that only local regions are edited, resulting in a masked image that remains consistent, the generated $z_0$, conditioned on the target prompt, is optimized to align with $\boldsymbol{y}$ to ensure background consistency.
>
> **Effects of Langevin Dynamics.** Since the applied method relies on a finite discrete scheme, there is a risk of becoming trapped in local optima. To mitigate this, Langevin dynamics is employed to search for the global optimal trajectory during gradient descent. Specifically, random perturbations are introduced in each gradient descent step. If the convergent point represents the global optimum, the iterative process will re-converge to this point even with the added perturbations. Conversely, if the point corresponds to a local optimum, the random perturbations assist the gradient descent process in overcoming the "U"-shaped region, facilitating continued exploration for the global optimum.
>
> > **W1(b).** More explanations for Eq. 16 about $\mathcal{A}$, m, h, and etc.
>
> **Forward Operator $\mathcal{A}$ and $n$.**  $\mathcal{A}$ represents the forward model, defined as $\mathcal{A}(z)=$\{0,1\}$^{n\times p}$, where $n$ and $p$ denote the dimensions of the measurement $y$ and the latent vector $z$, respectively. The distribution and number of 0 and 1 in the matrix \{0,1\}$^{n\times p}$ are randomized, enabling the generation of a random mask for each $z$. This measurement term is constructed as a combination of linear and nonlinear transformations of the initial image's background information, enabling robust background preservation.
>
>
> **Parameters $h$ and $m$.** $h$ is the learning rate. $h$ is firstly set to 1e-5 and then dynamically adjusted by different timesteps. The details are shown in the Eq.(20) of Appendix A.2. $m$ shown in the denominator is set to a value of 0.01.
>
> **Eq. (16).** This equation represents the core optimization process of PostEdit for the sampled $z_0$. Fundamentally, it is a gradient descent algorithm that iteratively updates $z_0$. The left-hand side of Eq. (16) denotes the updated $z_0$ after one iterative step. On the right-hand side, the first term corresponds to the optimized value from the previous step, The parameter $w$ is typically set to 0.1. The second term incorporates the input image, which is required to be edited according to the specified target prompt. The third term consists of the sum of two gradient components and. The first gradient term ensures that $z_0^{(k)}$ approaches the generated $z_0$ conditioned on the target prompt, aligning the optimization process with the desired editing objectives. The parameter $\sigma_t$, appearing in the denominator, corresponds to the DDPM scheduler as implemented in the HuggingFace library. The second gradient term is designed to align $z_0^{(k)}$ with the pre-defined measurement, thereby preserving background consistency. The final term represents a random perturbation, rendering the entire right-hand side of Eq. (16) equivalent to a sampling process governed by Langevin dynamics. Since the optimization process is implemented within a finite discrete scheme, the introduction of random perturbations ensures convergence to the global optimum while preventing stagnation in local optima.

---

> ### Author Response · Authors · 2024-11-22
> **Reply for Reviewer 98Bk 2**
>
> (To be continue) Last, all the parameters shown in Eq.(16) and other parts of PostEdit are shown specifically in Appendix A.2.
>
> > **W1 (c).** How to obtain the mask.
>
> We apply random masking to ${z}_0$ with a ratio of 50%. Specifically, we define a binary mask matrix \{0, 1\}$^{n \times p}$, which is element-wise multiplied with the latent $z$. Here, $n$ and $p$ represent the dimensions of the measurement $y$ and the latent vector $z$, respectively. The probability of an element being 0 in this mask matrix is set to 0.5. For the measurement $y$, random noise is added after applying the mask matrix to ${z}_0$. Further details can be found in Appendix A.2.
>
> Additionally, we conducted a parameter sensitivity analysis on the masking ratio. The results are summarized in the table below. A higher masking ratio results in better background preservation (higher reconstruction quality) but reduces the adherence to the editing prompt (lower Edited CLIP similarity). Our chosen setting of 50% achieves a balance between these objectives.
> | Masking Ratio | Background Preservation |         |         |         | CLIP Similarity |         | Efficiency |
> |---------------|--------------------------|---------|---------|---------|-----------------|---------|------------|
> |               | PSNR ↑                  | LPIPS ↓ ×10² | MSE ↓ ×10³ | SSIM ↑ ×10² | Whole ↑        | Edited ↑ | Time ↓    |
> | 30%           | 27.20                   | 6.09    | 2.91    | 82.77   | 25.93          | 22.40   | ~1.5s      |
> | 50% (used)    | 27.04                   | 6.38    | 3.24    | 82.20   | 26.76          | 24.14   | ~1.5s      |
> | 70%           | 24.43                   | 12.16   | 6.06    | 77.64   | 26.73          | 24.28   | ~1.5s      |
> > **W2(a).** Missing baselines. Add comparisons to SPD Inversion, and Guide-and-Rescale.
>
> Thanks! We quantitatively and qualitatively compared the mentioned SPD Inversion and Guide-and-Rescale. The results are presented in **Table 2**, **Figure 4** and **Figure 12**.
> - **Comparison with Guide-and-Rescale**:
>   PostEdit achieves superior performance across all quantitative metrics. This conclusion is further supported by qualitative results, as shown in **Figure 12**.
> - **Comparison with SPD Inversion**:
>   While PostEdit performs slightly worse in background preservation metrics (e.g., 1.82 dB lower PSNR), it offers significant advantages in CLIP similarity (resulting in higher editing quality), and efficiency (**20× faster**). Furthermore, as shown in **Figure 4**, PostEdit generates higher-quality images with better harmonization and stronger alignment to the edit prompts. Despite the minor difference in PSNR, PostEdit preserves the background effectively which is visually comparable to SPD Inversion.
>
> > **W2(b).** The ablation study lacks quantitative results to further support the analysis (e.g., posterior sampling vs. inversion process).
>
> Thanks! We have supplemented the quantitative results in **Table 5** of Appendix A.5. These results include the following ablation settings: (1) **No posterior sampling**, (2) **No mask**, and (3) **No $z_{in}$**.
>
> > **W2(c).** Add User Study to provide a more convincing evaluation.
>
> Thank you for the suggestion! The User Study is provided in **Appendix A.13**. We invited **34 anonymous volunteers** to their several preferred editing results obtained from different baselines. The feedback is shown detailed in Table 7 and 8.
>
>
> > **W2(d).** Provide reconstruction results for complex images containing multiple detailed objects.
>
> Thank you for the suggestion! In **Figure 3** (3rd row), we present reconstruction results for images containing multiple objects with occlusion relationships. Additional reconstruction results are provided in **Figures 15 and 16** of Appendix A.10. We encourage the reviewers to refer to these supplementary results to comprehensively assess the reconstruction performance of our method.

---

> > ### Comment · Reviewer_98Bk · 2024-11-26
> >
> > For the figure16, why the reconstruction results lack most of high-frequency details?

---

> ### Author Response · Authors · 2024-11-27
> **Reply for Reviewer 98Bk 3**
>
> Thanks for your insightful question! The reconstruction results shown in **Figure 16** are obtained using our default settings shown in Appendix A.2, which balance efficiency and reconstruction fidelity. However, as shown in **Figure 19** (newly submitted), **PostEdit can preserve more high-frequency details of the input image with additional optimization steps (e.g., 10,000 steps).**
>
> From **Figure 19**, we can observe that Null-Text Inversion (NTI), SPD Inversion, and our PostEdit achieve significantly better reconstruction results than other baselines. The possible reason is that incorporating information from the input image during optimization improves reconstruction accuracy.
>
> Notably, compared to NTI and SPD, which iteratively rectify the sampling trajectory, PostEdit employs posterior sampling explicitly conditioned on $x_0$, effectively addressing global errors in the sampling process. This enables PostEdit to achieve superior reconstruction results with higher PSNR (24.45 dB vs. 23.80 dB for SPD and 23.54 dB for NTI) and greater efficiency (15 seconds vs. 30 seconds for SPD and 120 seconds for NTI).

---

### Official Review · Reviewer_Psjc · 2024-11-03

**Soundness:** 3
**Presentation:** 3
**Contribution:** 3
**Rating:** 6
**Confidence:** 4

**Summary:**

This paper introduces a diffusion-based method designed for image reconstruction and editing, which operates within an encoded latent space. The proposed Langevin update rule for the latent vector at time $t+1$ consists of three components: a convex combination of the initial clean latent vector and the latent vector at time $t$, a negative gradient sum that guides the vector toward the initial noisy latent vector, and the measurement, and added Gaussian noise.

The authors evaluate the proposed method through image reconstruction and editing experiments, providing qualitative and quantitative comparisons with state-of-the-art approaches. In editing experiments, the method outperforms the competitors in CLIP Similarity and achieves competitive results in other metrics while also being one of the fastest methods.

**Strengths:**

The proposed algorithm is both inversion-free and training-free, contributing to its fast performance. The manuscript is well-written and clear. In editing experiments, the algorithm outperforms state-of-the-art approaches in CLIP similarity.

**Weaknesses:**

In image restoration experiments, quantitative evaluation is lacking, as the authors provide only qualitative comparisons with competing methods. Furthermore, the authors do not provide evidence that the reconstruction results produced by their method are consistent with the input measurements. Specifically, it would be helpful to see whether applying the forward measurement operator to the algorithms' output yields results that closely approximate the original measurements.

**Questions:**

- In equation 16, the gradient symbol is missing before the large brackets. The same issue occurs in line 13 of Algorithm 1 and row 710 in Algorithm 2.

- For a more comprehensive background, it may be beneficial to include a citation to [1], which introduced a method for solving linear image restoration problems in cases with noisy observations.

[1] Kawar, Bahjat, Gregory Vaksman, and Michael Elad. "SNIPS: Solving noisy inverse problems stochastically." Advances in Neural Information Processing Systems 34 (2021): 21757-21769.

---

> ### Author Response · Authors · 2024-11-22
> **Reply for Reviewer Psjc**
>
> **Thank you for your valuable review and suggestions. Below, we address each comment point-by-point.  We are looking forward to your further discussion and questions.**
>
> > **W1.** In image restoration experiments, quantitative evaluation is lacking.
>
> The qualitative results of images is shown in Table 1 in the revision. We compared PostEdit with five inversion-based methods specially designed for image reconstruction. The results in Table 1 demonstrate that PostEdit achieves a significant improvement in generation efficiency with minimal loss in generation quality. Although NTI slightly outperforms in image quality, it cost 120s, 10 times slower (120s vs 1.5s)
>
>
> > **W2.** The authors do not provide evidence that the reconstruction results produced by their method are consistent with the input measurements. Specifically, it would be helpful to see whether applying the forward measurement operator to the algorithms' output yields results that closely approximate the original measurements.
>
> Recall that there are two kinds of measurements introduced in our manuscript:
> 1. One (described in Eq. 21) incorporates additional information and is expected to achieve superior performance when focusing exclusively on reconstruction tasks.
> 2. Another (described in Eq. 17) is designed to balance reconstruction and editing capabilities.
>
> **All our image reconstruction results are generated using the measurement $y$ described in Eq. 17.** The reconstruction results of the two measurements are presented in **Figure 13** of Appendix A.8. Additionally, we supplemented more reconstruction results based on the used measurement (Eq. 17) in **Figure 15 and 16** of Appendix A.10. We encourage reviewers to refer to these figures for a comprehensive evaluation. The quantitative comparison results of these two forward operators are shown as follows:
>
> | Forward Operator | Background Preservation |         |         |         | Efficiency |
> |--------|--------------------------|---------|---------|---------|------------|
> |        | PSNR ↑                  | LPIPS ↓ ×10² | MSE ↓ ×10³ | SSIM ↑ ×10² | Time ↓    |
> | Ours (Eq.(21))   | 24.90                   | 7.60   | 4.31   | 74.03   | ~1.5s        |
> | Ours (Eq.(17))   | 24.39                   | 9.00    | 4.75    | 72.74   | ~1.5s | -->
>
> > **Q1.** In equation 16, the gradient symbol is missing before the large brackets. The same issue occurs in line 13 of Algorithm 1 and row 710 in Algorithm 2.
>
> Thanks! We have addressed the typos the reviewer identified. Please review the changes in the submitted PDF file.
>
> > **Q2.** Introducing "SNIPS: Solving noisy inverse problems stochastically." (In NeurIPS 2021) for a more comprehensive background.
>
> Thanks! We have highlighted this work in the Introduction and Method sections for readers better understanding inversion problems and noisy observation.

---

> > ### Comment · Reviewer_Psjc · 2024-11-26
> >
> > Thank you for the response. After carefully reviewing the feedback from other reviewers and considering the authors' rebuttal, I have decided to maintain my original rating.

---

> > > ### Author Response · Authors · 2024-11-27
> > > **Reply for Reviewer Psjc 2**
> > >
> > > Thanks for your careful and professional review! This has further improved the quality of our papers.

---

### Official Review · Reviewer_6fQq · 2024-11-04

**Soundness:** 3
**Presentation:** 2
**Contribution:** 3
**Rating:** 8
**Confidence:** 5

**Summary:**

This paper tackles the issues caused by the unconditional term in Classifier-Free Guidance (CFG) by integrating the theory of posterior sampling to improve reconstruction quality for image editing. By reducing the reliance on repeated network inference, the proposed method, PostEdit, achieves fast and accurate performance while successfully maintaining background consistency. The results seem promising.

**Strengths:**

1. By extending the theory of posterior sampling to text-guided image editing tasks, the proposed method, PostEdit, eliminates the need for both inversion and training.

2. PostEdit is one of the fastest zero-shot image editing approaches, achieving execution times of under 2 seconds on an A100 GPU.

**Weaknesses:**

While the paper presents an interesting approach, some details and experimental results are not sufficiently comprehensive. Key aspects of the methodology are not fully elaborated, and additional experiments would be beneficial to further validate the claims.

1. The subfigure "Our Posterior Sampling Process" in Fig. 1 is difficult to understand. It is unclear what exactly it is meant to represent. Additionally, how does it highlight the advantages of the proposed algorithm?

2. The authors mention that the proposed algorithm runs in less than 2 seconds, but it is not clear how many denoising steps are actually performed. In Algorithm 1, the values of N and n are not specified. These details are crucial because if the number of denoising steps is high, it could naturally lead to slower execution. It would also be beneficial to visualize some intermediate results for better clarity.

3. Regarding the editing process, how is the mask generated and provided? There is a lack of details on this aspect.

4. The current editing experiments only involve replacing a few words in the prompt, which seems quite limited. Does the method support more extensive editing operations? More explanation and additional experiments would be appreciated.

**Questions:**

Please see the Weaknesses part.

---

> ### Author Response · Authors · 2024-11-22
> **Reply for Reviewer 6fQq**
>
> **Thank you for your valuable review and suggestions. Below, we address each comment point-by-point. We are looking forward to your further discussion and questions** .
>
> > **W1.** Fig. 1 is hard to understand and the advantage of ours is unclear.
>
> Thanks! We have refined Figure 1 in the revised manuscript to better distinguish our method. It now more clearly highlight the introduced posterior sampling process, which reintegrates information from the initial image to ensure background consistency without additional training or inversion. We would greatly appreciate any further suggestions you might have for improvement.
>
> > **W2.** Details about the proposed Algorithm 1.
>
> > (1) It is not clear how many denoising steps are actually performed.
>
> We perform 5 denoising steps and use the LCM sampler to obtain a clean image, balancing efficiency and high generation quality. All hyperparameter settings are detailed in Appendix A.2.
>
> > (2) Specifying $n$ and $N$ in Algorithm 1.
>
> $N$ and $n$ are set to 5 and 1, respectively. We have updated Algorithm 1 in the revised manuscript to include these specifications.
>
> > (3) It would also be beneficial to visualize some intermediate results for better clarity.
>
> Intermediate results for $z_0$ and $z_t$ are displayed in the middle and bottom sections on the left-hand side of Figure 2. More detailed results are provided in Figure 14 of Appendix A.9, and please refer to it.
> > **W3.** Regarding the editing process, how is the mask generated and provided?
>
> We apply random masking to ${z}_0$ with a ratio of 50%. Specifically, we define a binary mask matrix \{0, 1\}$^{n \times p}$, which is element-wise multiplied with the latent $z$. Here, $n$ and $p$ represent the dimensions of the measurement $y$ and the latent vector $z$, respectively. The probability of an element being 0 in this mask matrix is set to 0.5. For the measurement $y$, random noise is added after applying the mask matrix to ${z}_0$. Further details can be found in Appendix A.2.
>
> Additionally, we conducted a parameter sensitivity analysis on the masking ratio. The results are summarized in the table below. A higher masking ratio results in better background preservation (higher reconstruction quality) but reduces the adherence to the editing prompt (lower Edited CLIP similarity). Our chosen setting of 50% achieves a balance between these objectives.
> |Masking Ratio        | Background Preservation       |         |         |         | CLIP Similarity      |         | Efficiency |
> |--------------|-------------------------------|---------|---------|---------|----------------------|---------|------------|
> |              | PSNR ↑                       | LPIPS ↓ ×10² | MSE ↓ ×10³ | SSIM ↑ ×10² | Whole ↑             | Edited ↑ | Time ↓    |
> | 0.3         | 27.20                        | 6.09    | 2.91    | 82.77   | 25.93               | 22.40   | ~1.5s      |
> | 0.7         | 24.43                        | 12.16    | 6.06    | 77.64   | 26.73               | 24.28  | ~1.5s       |
> | 0.5 (used)         | 27.04                        | 6.38    | 3.24    | 82.20   | 26.76           | 24.14| ~1.5s | -->
>
>
> > **W4.** The current editing experiments only involve replacing a few words in the prompt, which seems quite limited. Does the method support more extensive editing operations? More explanation and additional experiments would be appreciated.
>
> Since the presented method is both training- and tuning-free, PostEdit retains the expressive capabilities of the baseline text-to-image generation models. This enables it to effectively support adding, deleting, and replacing words in prompts, as demonstrated in Figure 4. For example, in the first row of Figure 4, additional words are appended to the end of the prompt, while the first three words are removed, preserving "flowers." These results highlight PostEdit's ability to perform extensive and diverse editing operations.
>
> Furthermore, we exhibit long-text editing effect of PostEdit: the original prompt and edit prompt both are long caption composed of several sentences, generated by GPT-4o. **The editing process includes not only replacing long sentences but also multiple deleting and adding operations**. The results, provided in Figure 8–11 in Appendix A.6 of the revised manuscript, confirm that PostEdit is not constrained by the number of words in the prompts and can effectively handle substantial modifications.
>
> Long-Text Editing Setup: We use GPT-4o to rewrite the original and edit prompts in our benchmark dataset (**PIE-Bench**) to be longer but describe the same content.

---

> > ### Comment · Reviewer_6fQq · 2024-11-29
> >
> > Thanks to the detailed clarification furnished by the authors and their painstaking efforts, after carefully perusing the response and taking into account the remarks of fellow reviewers, I've decided to upgrade my rating.

---

> > > ### Author Response · Authors · 2024-11-30
> > > **Reply for Reviewer 6fQq 2**
> > >
> > > We sincerely appreciate your professional and thorough comments, which have significantly contributed to the improvement of this paper.

---

### Author Response · Authors · 2024-11-22
**Summary of The Contents of Our Reply**

**We sincerely thank all reviewers for their time and effort in reviewing our paper. We are thrilled to see that the reviewers recognized the novelty of our contributions (Reviewers 6fQq, Psjc, 98Bk, CdSe) and appreciated the paper's clarity and theoretical depth (Reviewers Psjc, 98Bk). We are looking forward to your further discussion and questions.**

We also sincerely appreciate the reviewers for their professional, specific, and constructive comments and concerns. To address these, we have made modifications and provided detailed explanations in the attached PDF file for further clarification. Below, we summarize the key contents included in the attached file:

- **Figure 1 (to Reviewer 6fQq)** is revised for better comprehension of the presented three categories.
- **Algorithm 1 and 2 (to Reviewer Psjc)** are revised according to the reviewer's advice.
- **The operator $\mathcal{A}$ (to Reviewer 98Bk)** shown in Eq.(16) is illustrated and the content is following Eq.(16).
- **Citation [1] (to Reviewer Psjc)** is added in the introduction and method sections.
- **Figure 3 and Table 1 (to Reviewer 98Bk, CdSe)** are updated.
- **Appendix A. 3, Appendix A. 5, Appendix A. 6, Appendix A. 7, Appendix A. 8, Appendix A. 9, Appendix A. 10 and Appendix A. 12 (to Reviewer 6fQq, Psjc, 98Bk, CdSe)** have been added to show required experimental results.
- **A 'Related Work' section (to Reviewer CdSe)** is added in Appendix A.1.

Last, the presentation of the full text has been optimized to enhance clarity and improve understanding.

[1] Kawar, Bahjat, Gregory Vaksman, and Michael Elad. "SNIPS: Solving noisy inverse problems stochastically." Advances in Neural Information Processing Systems 34 (2021): 21757-21769.

---

> ### Author Response · Authors · 2024-11-27
> **Summary of The Contents of Our Reply 2**
>
> - **Figure 19 (to Reviewer 98Bk)** is added.

---

> > ### Author Response · Authors · 2024-11-28
> > **Summary of The Contents of Our Reply 3**
> >
> > We discovered that our previous image storage method led to a decrease in image resolution. This issue has now been resolved. Please feel free to check the updated **Figure 4** and **Figure 12 (to Reviewer 6fQq, Psjc, 98Bk, CdSe)** at your convenience. Thanks!

---

### Meta-Review · Area_Chair_Cw4V · 2024-12-16

**Metareview:**

This paper introduces PostEdit, a novel method for zero-shot image editing that leverages posterior sampling to achieve both efficiency and background preservation. The paper claims that PostEdit overcomes the limitations of existing inversion-based and inversion-free methods by incorporating a posterior scheme to guide the diffusion sampling process. The paper introduces a measurement term related to both initial features and Langevin dynamics to optimize the estimated image generated by the target prompt. The key finding is that PostEdit achieves state-of-the-art editing performance while preserving unedited regions with high efficiency.

Strengths:
- Efficiency: PostEdit is both inversion- and training-free, requiring only 1.5 seconds and 18 GB of GPU memory for high-quality results.
- Background Preservation: The method effectively preserves unedited regions of the image, addressing a common challenge in image editing.
- Theoretical Foundation: The incorporation of posterior sampling and Langevin dynamics provides a solid mathematical framework for the approach.
- Clarity and Depth: The paper is well-written and presents the methodology with clarity and theoretical depth.

Weaknesses:
- Limited Editing Operations: The initial experiments primarily focused on replacing a few words in the prompt, potentially limiting the scope of editing operations.
- Lack of Complex Image Reconstruction: The reconstruction results mainly involved images with single objects and simple layouts, lacking demonstrations with complex images containing multiple detailed objects.
- Missing Baselines: The initial experimental comparisons lacked some state-of-the-art methods, such as SPD Inversion and Guide-and-Rescale.
- Limited Ablation Study: The initial ablation study lacked quantitative results to fully support the analysis of individual components' effectiveness.

Despite the identified weaknesses, the paper presents a novel and efficient approach to zero-shot image editing with nice theoretical formulation. The method addresses important challenges in the field, particularly in terms of efficiency and background preservation. The authors have also diligently addressed the reviewers' concerns during the rebuttal period by providing additional experiments, clarifications, and analyses. The overall consensus among the reviewers, coupled with the paper's strengths and the authors' responsiveness, supports the decision to accept.

**Additional Comments On Reviewer Discussion:**

The reviewers raised several pertinent points during the discussion, including the clarity of Figure 1, the details of Algorithm 1, the generation of the mask, the range of editing operations supported, the comprehensiveness of the baselines, the need for quantitative results in the ablation study, the inclusion of a user study, and the demonstration of complex image reconstruction. The authors diligently addressed each of these points by providing detailed clarifications, revising the figures and algorithms, and conducting additional experiments. Their responsiveness and the thoroughness of their responses significantly contributed to the final decision to accept the paper.

---

### Decision · Program_Chairs · 2025-01-22

Accept (Poster)